# Representativeness of individual-level data in COVID-19 phone surveys: Findings from Sub-Saharan Africa

Joshua Brubaker[1], Talip Kilic [1], Philip Wollburg [2]*

1 Living Standards Measurement Study, Development Data Group, World Bank, Washington, D.C., United States of America, 2 Living Standards Measurement Study, Development Data Group, World Bank, Rome, Italy

* pwollburg@worldbank.org

## Abstract

The COVID-19 pandemic has created urgent demand for timely data, leading to a surge in mobile phone surveys for tracking the impacts of and responses to the pandemic. Using data from national phone surveys implemented in Ethiopia, Malawi, Nigeria and Uganda during the pandemic and the pre-COVID-19 national face-to-face surveys that served as the sampling frames for the phone surveys, this paper documents selection the biases in individual-level analyses based on phone survey data. In most cases, individual-level data are available only for phone survey respondents, who we find are more likely to be household heads or their spouses and non-farm enterprise owners, and on average, are older and better educated vis-a-vis the general adult population. These differences are the result of uneven access to mobile phones in the population and the way that phone survey respondents are selected. To improve the representativeness of individual-level analysis using phone survey data, we recalibrate the phone survey sampling weights based on propensity score adjustments that are derived from a model of an individual's likelihood of being interviewed as a function of individual- and household-level attributes. We find that reweighting improves the representativeness of the estimates for phone survey respondents, moving them closer to those of the general adult population. This holds for both women and men and for a range of demographic, education, and labor market outcomes. However, reweighting increases the variance of the estimates and, in most cases, fails to overcome selection biases. This indicates limitations to deriving representative individual-level estimates from phone survey data. Obtaining reliable data on men and women through future phone surveys will require random selection of adult interviewees within sampled households.

## 1. Introduction

With the onset of the coronavirus disease 2019 (COVID-19) pandemic, governments, academic institutions, and international organizations scrambled to measure and monitor the pandemic's impacts on livelihoods and tailor policy responses. A global survey of National

**Data Availability Statement:** The data are available from the World Bank's Microdata Library, High Frequency Phone Survey Catalog (https://microdata.worldbank.org/index.php/catalog/hfps)

as well as LSMS catalog (https://microdata.
worldbank.org/index.php/catalog/lsms).

**Funding:** Funding for data collection and analysis
comes from the World Bank Multi-Donor Trust
Fund for Integrated Household and Agricultural
Surveys in Low and Middle-Income Countries
(TF072496). The funders had no role in study
design, data collection and analysis, decision to
publish, or preparation of the manuscript.

**Competing interests:** The authors have declared
that no competing interests exist.

Statistical Offices (NSOs) showed that already in May 2020 over 80 percent were involved in collecting data related to the COVID-19 pandemic, focusing predominantly on its socioeconomic and business impacts. However, prompted by lockdowns, travel restrictions and safety concerns, face-to-face (F2F) survey data collection was suspended in most countries at the onset of the pandemic. Since then, the movement to resume F2F surveys, even under strict COVID-19 fieldwork protocols, has been slow and uncertainty regarding the timeline for fully resuming activities under the "new normal" prevails [1]. These developments have led to a proliferation of telephone surveys as the tool of choice for collecting data on COVID-19 impacts among the majority of NSOs [2, 3]. Similarly, the World Bank has launched a global initiative to monitor COVID-19 impacts using phone surveys as have UN Women, Innovations for Poverty Action and Young Lives, among many others [4–7]. In the meantime, insights derived from these phone surveys have been used widely in published research and to inform policy.

Phone surveys with national coverage had previously been rather uncommon in low-income countries and relatively little was known about their feasibility and best practices. However, the COVID-19 pandemic accelerated a more widespread adoption of phone surveys as an instrument of choice in low-income countries, so that phone surveys are likely to remain commonplace even after the COVID-19 pandemic, complementing F2F surveys [8].

Making effective use of phone survey data for research and to inform policies in low-income countries now and in the future requires addressing selection biases from which phone surveys are more prone to suffer than F2F surveys and which threaten the representativeness of estimates based on phone survey data.

In this paper, we document how selection biases affect individual-level data derived from phone surveys from four Sub-Saharan African countries and assess to what extent these biases can be addressed through reweighting. Individual-level analysis is of special interest in this context because important outcomes such as attitudes towards and knowledge of the COVID-19 pandemic are individual outcomes that are captured for the survey respondent only. For example, a recent study uses individual respondent level phone survey data to examine the acceptance of COVID-19 vaccines in six Sub-Saharan African countries [9]. Moreover, individual-level data are critical to properly understand the heterogenous impacts of COVID-19 by gender, age group, and other subpopulations of interest [10, 11]. Since phone ownership is less common among women and vulnerable populations, surveying these groups in a representative fashion is a particular challenge, especially in the context of the COVID-19 pandemic [12, 13].

Phone surveys are prone to various forms of selection biases. First, phone surveys usually require phone ownership; in low-income countries phone ownership is not universal, which may lead to coverage bias. A review paper of 15 phone-based studies from 11 low- and middle-income countries finds that phone survey samples are skewed towards men and individuals in wealthier, male-headed, urban and better-educated households and therefore under-represent certain parts of the population [14]. Second, response rates in phone surveys are lower than in F2F surveys because respondents do not pick up, refuse participation at higher rates, or phone numbers are disconnected. This leads to non-response bias when responding households are systematically different from households that do not respond. A recent study documents the nature and extent of both coverage and non-response biases at the household level in phone survey data from Ethiopia, Malawi, Nigeria, and Uganda [15]. The study finds that households in phone survey samples are wealthier and less likely rural or agricultural than a nationally representative sample of households. A pre-COVID-19 study find similar patterns in phone surveys in South Sudan, Tanzania, and Honduras [16].

For individual-level data, which we focus on in this paper, a further potential source of bias is related to respondent selection. Most phone surveys are done with just one main respondent and respondent selection protocols often target heads of households or "most knowledgeable" adult household members such that the sample of respondents may not be representative of the general adult population. Selecting the "most knowledgeable" adult as a respondent is a common practice in household surveys, whether face-to-face or telephone, and concerns with individual-level representativeness arise from this choice in all cases. However, interviewing household members other than the main respondent, or asking the main respondent to report information on behalf of other household members, is considerably easier and more common in F2F surveys [17, 18].

The severity of these biases may vary depending on the phone survey mode [19, 20] and depending on the sampling strategy [14]. Three main sampling strategies have been employed for phone surveys in low- and middle-income countries, both during the COVID-19 emergency and before. First, based on phone numbers collected in a previous F2F survey. Second, using a list of phone numbers otherwise obtained, for example from a mobile network operator. Third, random digit dialing (RDD), whereby randomly generated phone numbers are called, which is used widely when no pre-existing list of phone numbers is available [21]. Phone surveys based on RDD in low- and middle-income countries have been found to suffer from significantly higher non-response rates than those based on existing contact information, which may in turn lead to greater non-response bias [14, 21]. However, RDD-based phone surveys typically use individual phone numbers and may face less of a respondent selection problem.

The advantage of a sampling strategy based on an existing list of phone numbers from a representative F2F survey is that there is a wealth of information on each household or individual with a phone number as well as on households or individuals without a phone number. This information in turn can be used to characterize selection biases and recalibrate sampling weights to improve the representativeness of the phone survey data–a feature that we will also make use of in this analysis. A recent study using phone survey data from Ethiopia, Malawi, Nigeria, and Uganda shows that recalibrating survey weights is relatively successful at overcoming coverage and non-response biases at the household-level [15]. Reweighting was also used to improve the representativeness of a study on the impacts of the Ebola crisis in Liberia and Sierra Leone, albeit without a systematic attempt at assessing the relative success of this method [22].

Our analysis leverages data from national high-frequency phone surveys on COVID-19 (HFPS) in Ethiopia, Malawi, Nigeria and Uganda and the nationally-representative F2F surveys that had been implemented prior to the pandemic under the World Bank Living Standards Measurement Study–Integrated Surveys on Agriculture (LSMS-ISA) program and that served as the sampling frames for the phone surveys. The F2F surveys collected the phone numbers of at least one individual per household, and in some cases of all household members, which were then used to contact households for the high-frequency phone surveys. This setup allows us to compare phone survey respondents and the general adult population along a range of individual and household characteristics.

Our analysis confirms that concerns regarding the representativeness of individual-level phone survey data are warranted. Selected phone survey respondents are most often household heads or their spouses, and on average, are older, better educated and more likely to own a non-farm enterprise vis-a-vis the general adult population. To account for these differences and improve the representativeness of individual-level phone survey data, we recalibrate the household-level phone survey sampling weights based on propensity score adjustments that are derived from a cross-country comparable model of an adult individual's likelihood of

being interviewed in a phone survey household as a function of a rich set of individual- and household-level attributes [23] and assess to what extent the recalibrated weights can address selection biases. Reweighting generally improves the representativeness of the individual-level estimates, moving the variable means for phone survey respondents closer to those of the general adult population. This holds for both women and men and for a range of demographic, education, and labor market outcomes. However, reweighting increases the variance of the estimates and fails to fully overcome individual-level selection biases, with differences in means remaining statistically significant for the majority of outcomes–somewhat contrary to what a recent study with the same data sources found for household-level biases [15]. Obtaining reliable individual-level data from these phone surveys, therefore, requires fundamental changes to the individual respondent selection protocols with a focus on random selection of interviewees.

Our paper is part of a growing literature on methodology and best practices for designing and conducting phone surveys in low- and middle-income countries, covering a range of issues including sampling [21, 24]; survey mode [14, 20, 25]; survey cost, non-response, attrition, and use of incentives [16, 26–31]; and questionnaire design [19, 32, 33]. There are also several guidebooks and synthesis reports that summarize best practices and experiences with phone surveys from before the COVID-19 pandemic [16, 26, 29] as well as in the context of the COVID-19 pandemic [8, 14, 32].

The remainder of the paper is structured as follows. Section 2 describes the data and methods we use to assess individual-level biases and the relative success of bias reduction techniques. Section 3 presents the main emerging findings. Section 4 concludes with a discussion of what the results mean for individual level analysis and data collection using phone surveys.

## 2. Data and methods

### 2.1. Data sources

The longitudinal survey data informing our analysis originate from (i) the national high-frequency phone survey (HFPS) that was implemented on a monthly basis in Ethiopia, Malawi, Nigeria and Uganda during the COVID-19 pandemic, and (ii) the pre-COVID-19 F2F household survey that served as a sampling frame for each HFPS.

Each pre-COVID-19 F2F survey that was the source of the phone numbers for the respective country had been designed to be representative at the national, regional, and urban/rural levels. These F2F surveys are the Ethiopia Socioeconomic Survey (ESS) 2018/19, the Malawi Integrated Household Panel Survey (IHPS) 2019, the Nigeria General Household Survey (GHS)—Panel 2018/19, and the Uganda National Panel Survey (UNPS) 2019/20. In Ethiopia, Malawi, and Uganda, the HFPS attempted to call all pre-COVID-19 F2F survey households for which at least one phone number was available. The Nigeria HFPS first drew a national sub-sample from the universe of F2F survey households with contact details, based on a balanced sampling approach using the cube method [34], before this sub-sample of households was contacted.

In Ethiopia, we use data from the first round of the HFPS, which was implemented in April-May 2020, covering 3,249 households. In Malawi, we use data from the first and fifth rounds of the HFPS, which were implemented in May-June 2020 and October-November 2020, covering 1,729 and 1,589 households, respectively. Similarly, in Nigeria, we use data from the first and fifth rounds of the HFPS, which were implemented in April-May 2020 and September 2020, covering 1,950 and 1,774 households, respectively. We use the fifth round of the HFPS in the specific cases of Malawi and Nigeria to analyze individual-level employment data which was collected on all adults in each household only in these two countries. Lastly, in

Uganda, we use data from the first round of the HFPS, which was implemented in June 2020, covering 2,227 households.

The implementing agency for the national phone surveys in Ethiopia, Malawi, Nigeria and Uganda are, respectively, Laterite Ethiopia, the Malawi National Statistical Office, the Nigeria Bureau of Statistics, and the Uganda Bureau of Statistics. The anonymized, unit-record phone survey data are available publicly through the World Bank Microdata Library under the High-Frequency Phone Survey collection [35]. The World Bank World Bank Microdata Library is the preferred platform for public dissemination among the NSOs in Ethiopia, Malawi, Nigeria, and Uganda. The approach to the phone survey questionnaire design and sampling was generally comparable across countries, albeit with some scope for contextualization, informed by a set of tools designed for the HFPS on COVID-19, including a template questionnaire, phone survey sampling guidelines, and computer-assisted telephone interviewing (CATI) guidelines [21, 33, 36, 37]. The template questionnaire included a set of core modules which were adopted across countries as well as a set of other modules which countries adopted optionally according to interest and need.

Since the phone surveys build on the F2F surveys, and the phone survey respondent was recorded using unique anonymized household and household member identification numbers, we can link the phone survey data with the pre-COVID-19 F2F survey data at the individual-level. This gives us two samples to compare: (i) the phone survey respondents and (ii) the general adult population, derived from the nationally representative pre-COVID-19 F2F sample of which the phone survey populations are a subsample. Individuals 15 and above were considered part of the general adult population as these individuals were eligible to be respondents in the HFPS and the F2F surveys.

Our analysis assesses the differences between phone survey respondents and the general adult population as represented in the pre-COVID-19 F2F surveys and gauges the success in utilizing bias correction techniques to derive general adult population representative estimates for a core set of individual-level variables related to gender, age, marital status, relationship with the household head, education, and employment. S1 Table shows the unweighted means of these variables for the samples of interest.

## 2.2. Ethics approval

Informed consent was received from all phone survey and F2F survey respondents in each country. The World Bank does not require institutional ethics approval for household surveys that are partly or fully financed by the World Bank, including the national phone surveys in Ethiopia, Malawi, Nigeria, and Uganda that inform our research. Furthermore, each phone survey was implemented by the respective national statistical office (NSO), except for Ethiopia where a private firm was the implementing agency. This means that in the specific cases of Malawi, Nigeria, and Uganda, the NSO conducts the survey as the sole official statistical authority in the country and in accordance with the respective National Statistical Act, which exempts the NSO from institutional ethics approvals. All data sets used were fully anonymized prior to our access, that is, all personal identifying information on households and individuals was removed and households and individuals were given anonymized identification numbers.

## 2.3. Sampling frames, contact protocols, and respondent selection

Though informed by the same general guidelines [36], the protocols for contacting the sampled households and subsequently selecting the respondent in each household were slightly different in each HFPS, reflecting country-level survey design choices as well as differences in how phone numbers were recorded in the pre-COVID-19 F2F surveys which served as

sampling frames. In Malawi, the IHPS 2019 was the sampling frame for the HFPS. During the IHPS 2019, phone numbers were collected from the sampled households in two ways: First, each household member's phone number was collected during the interview and recorded as part of the household roster, provided that the individual had a phone number. Second, phone numbers for up to three non-household reference contacts, such as neighbors or friends, were noted at the beginning of the interview. Prior to the implementation of the first round of the HFPS, the resulting list of phone numbers for each household was put in random order. During the first round of the HFPS, enumerators then called the phone numbers in accordance with this order in each household. However, the first contact was not necessarily the same person as the main respondent, since being the main respondent required an ability and willingness to respond to survey questions and thus it was possible for first contacts to hand over the phone to another person. In the following rounds, the first phone number to be called was the one that the respondent of the first round indicated as the best number to reach them. The original list of phone numbers was retained in the event that the preferred phone number could not be reached. Of the 3,181 IHPS 2019 households that were interviewed face-to-face, 2,337 provided at least one phone number and all of these households were attempted to be contacted by the HFPS. Of the attempted households, 1,729 households were fully interviewed in the first round, a response rate of 74 percent.

In Ethiopia, the ESS 2018/19 was the sampling frame for the HFPS. The ESS 2018/19 interviewed 6,770 households which were asked to provide phone numbers for the head of household, up to three additional household members and up to two non-household reference individuals. At least one phone number was obtained for 5,374 ESS 2018/19 households. The enumerators called the available phone numbers for each household in the order in which they were recorded during the ESS 2018/19 interview. During the first round of the HFPS, all 5,374 households were attempted to be contacted, of whom 3,249 were successfully interviewed, for a final response rate of 60 percent.

In Nigeria, the GHS-Panel 2018/19 was the sampling frame for the HFPS. The GHS-Panel 2018/19 interviewed 4,976 households of whom 4,934 provided phone numbers and from which 3,000 were in turn randomly selected to be contacted in the first round of the HFPS. The contact protocol targeted the household head, who was called first if their number was listed, followed by the remaining household members and the reference contacts in the order in which they were captured by the GHS-Panel 2018/19. During the first round of the HFPS, 1,950 households were successfully interviewed out of 3,000 households attempted, equivalent to a 65 percent response rate.

Finally, in Uganda, the UNPS 2019/20 was the sampling frame for the HFPS. The UNPS 2019/20 interviewed 3,098 households, of whom 2,386 provided a phone number for at least one household member or a reference contact. The HFPS attempted to contact all 2,386 households, of whom 2,227 were successfully interviewed, markedly the highest response rate in our sample at 93 percent. Like Nigeria, the Uganda HFPS contact protocol prioritized the household head, followed by other household members, and referenced contacts, in the order in which they were captured during the UNPS 2019/20.

Table 1 presents a summary of the sampling steps and pertinent sample sizes of the four HFPS used in this paper.

Across all households in the F2F survey database, there are a total of 17,563 adults in Ethiopia, 8,588 in Malawi, 15,230 in Nigeria, and 8,763 in Uganda–irrespective of being contacted or interviewed in one of the HFPS rounds that are used in our analysis. Of these adults, 8,004 in Ethiopia, 4,670 in Malawi, 6,178 in Nigeria, and 6,361 belonged to F2F survey households that were also interviewed in the first round of the HFPS.

**Table 1. Selection of HFPS households.**

| Sample Households (HHs) | Ethiopia N | Ethiopia % | Malawi N | Malawi % | Nigeria N | Nigeria % | Uganda N | Uganda % |
|---|---|---|---|---|---|---|---|---|
| Face-to-face (F2F) HH sample | 6,770 | 100 | 3,181 | 100 | 4,976 | 100 | 3,098 | 100 |
| HHs with phone numbers | 5,374 | 79.4 | 2,337 | 73.5 | 4,934 | 99.2 | 2,386 | 77.0 |
| HHs called by HFPS | 5,374 | 79.4 | 2,337 | 73.5 | 3,000 | 60.3 | 2,386 | 77.0 |
| HHs reached by HFPS | 3,357 | 49.6 | 1,743 | 54.8 | 2,057 | 41.3 | 2,246 | 72.5 |
| HHs successfully interviewed by HFPS | 3,249 | 48.0 | 1,729 | 54.4 | 1,950 | 39.2 | 2,227 | 71.9 |
| HHs successfully interviewed by HFPS with the phone survey respondent also appearing in the F2F survey | 3,196 | 47.2 | 1,701 | 53.5 | 1,910 | 38.4 | 2,128 | 68.7 |

Table 2 presents unweighted descriptive statistics for (i) individuals that were respondents in successfully interviewed HFPS households in round 1 (i.e. phone survey respondents), and (ii) all adults living in F2F survey households, irrespective of being contacted or interviewed by the HFPS (i.e. the general adult population). In all HFPS rounds that inform our analysis, the majority of respondents were household heads, ranging from 74 percent in Uganda to 83 percent in Ethiopia with Malawi and Nigeria standing at 79 and 82 percent, respectively. This similarity in the share of household heads interviewed across countries is notable because Ethiopia, Nigeria, and Uganda implicitly or explicitly targeted the household head as the HFPS respondent, whereas in Malawi the order of the contacted phone numbers was randomized for each household. One reason for this is that household heads are likely to own phones and as a result are more likely to be called. Another conceivable reason is that individual phone owners other than the household head handed phones to the household head to respond on behalf of the household. Next to household heads, in each country the remaining HFPS respondents were predominantly spouses of the household head.

A majority among phone survey respondents was male, ranging from 73 percent in Nigeria to just slightly above the population average in Uganda at 52 percent with Ethiopia and Malawi standing at 62 and 63 percent, respectively. The HFPS respondents were also much less likely to be among the youth (i.e. between the ages of 15 and 24 years) vis-à-vis the general adult

**Table 2. Unweighted descriptive statistics for HFPS respondents and adult population in F2F survey.**

| | | Ethiopia Phone resp. | Ethiopia Adult pop. | Malawi Phone resp. | Malawi Adult pop. | Nigeria Phone resp. | Nigeria Adult pop. | Uganda Phone resp. | Uganda Adult pop. |
|---|---|---|---|---|---|---|---|---|---|
| Gender | Women | 37.6 | 52.7 | 36.9 | 52.4 | 27.2 | 51.7 | 48.3 | 51.8 |
| | Men | 62.4 | 47.3 | 63.1 | 47.6 | 72.8 | 48.3 | 51.7 | 48.2 |
| Age Group | 15–24 | 12.9 | 34.3 | 11.8 | 39.6 | 5.7 | 31.6 | 5.9 | 37.7 |
| | 25–49 | 66.6 | 49.6 | 65.5 | 44.5 | 55.0 | 45.0 | 59.8 | 40.8 |
| | 50+ | 20.5 | 16.1 | 22.6 | 15.9 | 39.3 | 23.4 | 34.3 | 21.5 |
| Relationship to HH Head | Head | 82.8 | 38.5 | 78.7 | 37.0 | 82.7 | 32.7 | 74.1 | 35.1 |
| | Spouse | 9.8 | 24.8 | 16.5 | 26.1 | 9.2 | 28.1 | 20.2 | 22.0 |
| | Child | 6.0 | 26.3 | 3.1 | 24.6 | 6.5 | 30.3 | 4.4 | 32.1 |
| | Other | 1.5 | 10.3 | 1.8 | 12.3 | 1.7 | 9.0 | 1.4 | 10.8 |
| Observations | | 3,196 | 17,563 | 1701 | 8,588 | 1910 | 15,230 | 2128 | 8,763 |

**Note:** Table 2 presents unweighted results. Phone resp. = phone survey respondents; Adult pop. = General adult population as captured in pre-COVID-19 nationally representative household surveys. The sample underlying the estimates in this table exclude individuals that were HFPS respondents but that were not household members at the time of the pre-COVID19 F2F surveys. In Ethiopia, 98.4 percent of successfully interviewed households in the first HFPS round had a respondent that was also present in the associated F2F survey. This rate was 98.3 percent in Malawi, 97.9 percent in Nigeria, and 93.9 percent in Uganda.

population. The gap was most pronounced in Uganda where 6 percent of respondents versus 38 percent of adults fall in the 15–24 age range and was smallest in Ethiopia where 13 percent of respondents versus 34 percent of adults fall in the same age range. This finding is somewhat contrary to previous studies, which often found youth to be overrepresented among phone survey respondents [14].

## 2.4. Household and individual sampling weights

There are several sampling weights that are used in our analysis. To start with, there are the pre-COVID-19 F2F household survey sampling weights (**wb**). These sampling weights serve as the starting point for the computation of the HFPS household sampling weights in public use datasets (**w1**), which are calibrated versions of *wb* that address coverage and non-response biases at the household-level by leveraging the rich, pre-COVID-19 F2F survey data on (i) households that do not own a mobile phone and are excluded from the sampling frame; (ii) households that participate in the HFPS, and (iii) households that are contacted but cannot not be reached. This latter scenario is overwhelmingly due to non-working phone numbers or prospective respondents not answering calls as opposed to answering the phone call but then refusing to respond to the survey.

The household-level bias adjustment to create *w1* follows the methodology proposed in a previous methodological contribution [23] and detailed specifically for the HFPS rounds in Ethiopia, Malawi, Nigeria, and Uganda in a recent paper [38]. This methodology is also commonly used for the computation of sampling weights in longitudinal F2F surveys with tracking of individuals over time. The HFPS household sampling weights are further post-stratified to match the projected population totals at the highest spatial resolution possible, ranging from region to district, based on the data availability in each country.

Yet, *w1* does not account for the non-random selection of an *individual* to be a HFPS respondent. To address this and allow for the analysis of individual-level phone survey data in a way that is more representative of the general adult population, an additional individual-level sampling weight is needed. The objective of this paper is to assess the effectiveness of this recalibrated weight to correct for selection biases at the individual level. In what follows, we detail an approach that can be followed by any potential data user, leveraging solely the publicly available data on successfully interviewed HFPS households and their adult household members—as captured in the pre-COVID-19 F2F surveys and the HFPS.

To create the individual-level weight (**w2**), we follow an adjustment procedure that is similar to the procedure used to create *w1*. First, using the sample of all adult members of HFPS households (respondents and non-respondents), we estimate an unweighted logit regression to model the individual-level probability of selection as a HFPS respondent:

$$Pr(respondent = 1) = F\left(\beta_0 + \sum_{k=1}^{K} \beta_k X_k\right) \tag{1}$$

The dependent variable in this model is a binary variable indicating whether a given individual was the round 1 HFPS respondent. X is a vector containing K independent variables that originate from the F2F survey and that are expected to predict the likelihood of being a HFPS respondent. The sample for Eq 1 is individuals who were household members both in the pre-COVID-19 F2F surveys and in the HFPS. A cross-country consistent set of independent variables is used for Eq 1, including an extensive range of individual and household attributes and spatial fixed effects. Eq 1 is then estimated separately for each country. Since the individual's relationship to the household head is likely to impact respondent-ship due to the HFPS respondent selection protocols, dichotomous variables are included to identify household

head, spouse of the household, and child/adopted child of the household head, with the omitted category being any other relationship to the household head.

Additional dichotomous variables are included to identify (i) men; (ii) married individuals; (iii) those aged 25–49 and, separately, 50+, with individuals in the age range of 15–24 constituting the omitted category; (iv) individuals with completed primary education, completed secondary education, completed post-secondary certificate/training, and completed post-secondary degree, with individuals having less than completed primary education being the omitted category; and (v) individuals that can read and write in any language. Since individuals with different time use may have different incentives and availability to respond to a phone survey, a set of non-exclusive dichotomous variables are included to discern whether the individual had regular wage employment; was the owner of a household enterprise and participated in casual labor (with the latter being restricted only to Ethiopia and Malawi, in view of data availability and importance of casual labor activities in these contexts). Finally, a dichotomous variable is included to identify an individual's ownership of a mobile phone, which is expected to increase likelihood of being a HFPS respondent. The household-level attributes in Eq 1 are (i) household size, which is expected to decrease the probability of any single adult being a HFPS respondent; and (ii) dichotomous variables identifying the household's total annual per capita household consumption expenditure quintile, with the lowest quintile being the excluded category.

The significance level and size of the marginal effects associated with the regression coefficients ($\beta$) of the binary independent variables can be interpreted as the change in likelihood of being a phone survey respondent as a result of having the respective individual characteristic. Following the estimation of Eq 1, we predict the probability of being a HFPS respondent across the entire sample of adult household members in successfully interviewed HFPS households. Following guidance from the relevant literature [21, 23, 39], we then create deciles for this variable compute the average predicted probability within each decile, and take the reciprocal of this average to define the adjustment factor for each decile ($af_{D = d}$):

$$af_{D=d} = \frac{1}{\frac{\sum_{i=1}^{N} \widehat{respondent_i}}{N}} \tag{2}$$

where N is the number of individuals in each decile. The computation of the average probability per decile ensures that there are both respondents and non-respondents assigned to each value of the reweighting adjustment factor, creating covariate balance between respondents and non-respondents which the raw probability variable could not achieve [39]. The adjustment factor is then applied to $w1$, the HFPS household sampling weight in the public use phone survey dataset:

$$w_{i,af} = af_{D=d} * w1 \tag{3}$$

$w_{i,af}$ is in turn winsorized at the top and bottom 2 percent, in order to deal with extreme outliers, which reduces standard errors and makes estimates more efficient [23]. The winsorized weight is then post-stratified to equal population totals at the highest spatial resolution available, following the approach to the post-stratification of $w1$. Post-stratification ensures the weights sum up to known population totals and also reduces overall standard errors [23, 40]. In each country, the post-stratification adjustment ($w_{ps}$) is produced at the level of the lowest administrative unit for which population projections are available (typically region or district, depending on the country). It is computed as (i) the weighted total number of households residing in each administrative unit of interest, as measured by the sum of winsorized $w_{i,af}$ values in that unit, divided by (ii) the household population projection in that unit. Once

computed, $w_{ps}$ for each administrative unit is associated with all surveyed households in that unit, and $w_{i,af}$ is multiplied with $w_{ps}$ to derive the final individual weight, $w2$:

$$w2 = w_{ps} * w_{i,af} \tag{4}$$

## 2.5. Assessing differences between HFPS respondents and general adult population under different sampling weights

To assess the effectiveness of the bias reduction techniques for the individual-level phone survey data analysis, we focus on the individual-level variables that are captured in the pre-COVID-19 F2F survey and that are related to gender, age, marital status, relationship with the household head, education, and employment (see S1 and S2 Tables), which are the individual-level variables included in the logit regression as part of creating the recalibrated weight $w2$ (see section 2.4). We derive estimates of the mean of these variables using two different samples: (i) *all adult household members*, *as captured in the F2F survey*, who are assumed to be representative of the general adult population with the use of F2F household sampling weights (*wb*), and (ii) *HFPS respondents* who were also present in the F2F survey (i.e. ii is a subsample of i).

The weighted estimates for the adult household members in the F2F survey, denoted as *b*, serve as the benchmark to which we compare the sample of HFPS respondents under three different scenarios:

1. unweighted (*w0*),

2. weighted by the HFPS *household* sampling weights in the public use datasets (*w1*), and,

3. weighted by our newly generated HFPS *individual* sampling weight (*w2*), which is the recalibrated version of *w1*, intended to account for the non-random selection to be a HFPS respondent among the adult household members residing in the successfully interviewed HFPS households.

We use two different approaches to assess the effectiveness of HFPS household and individual sampling weights in reducing the bias in estimates for the HFPS respondents vis-à-vis the general adult population (as captured through the F2F survey). First, we take a graphical approach, where the estimates from the F2F and phone surveys are standardized by subtracting the F2F survey mean. This means that the F2F survey mean is always zero and all other estimates are standardized in relation to the F2F survey mean, allowing a comparison across the competing estimates. The graphs then present the weighted mean and 95 percent confidence interval estimated for a range of individual-level variables for the general adult population (*b*), and the same set of statistics estimated for the HFPS respondents, without the use of any sampling weight (*w0*) and employing the HFPS household (*w1*) or individual (*w2*) sampling weight. This allows us to assess how large the differences in the two populations are at the outset (*b* vs *w0*) and how well the HFPS household sampling weights (*b* vs *w1*), and the HFPS individual weights (*b* vs *w2*) perform in reducing the differences between the HFPS respondents and the general adult population.

Second, we rely on Wald tests to assess whether the HFPS-based estimates obtained under different weights are significantly different vis-à-vis the F2F survey-based estimates for the general adult population. This approach requires constructing an appended dataset containing:

i. all adult household members in the F2F survey households and the F2F survey household sampling weight (*wb*),

ii. the HFPS respondents and the HFPS household sampling weight set to 1 (*w0*),

iii. the HFPS respondents and the HFPS household sampling weight in the public use datasets (*w1*), and

iv. the HFPS respondents with the HFPS individual sampling weight (*w2*).

In this set up, the samples (ii) through (iv) are composed of identical individuals that are appended with different sampling weights and that constitute a subset of sample (i). A common name is used for the sampling weight variable across the appended datasets and each appended dataset includes the same set of individual-level variables, as listed in S1 and S2 Tables. Furthermore, a new categorical variable is defined to uniquely identify each appended sample (i through iv). A weighted linear regression is then estimated for each outcome of interest, with an identical set of independent variables that include the dichotomous variables identifying the samples (ii) through (iv), with the sample (i) (i.e., all adult household members in the F2F survey households) serving as the comparison category. The sampling weight for each observation is equal either to *wb*, *w0*, *w1* or *w2* in accordance with the sample that the record belongs to. When presenting the results from this regression, the base category is shown on the top row and represents the mean from which all other estimates deviate. The values in rows other than the base category express the difference in mean from the base category.

## 2.6. An application with phone survey data

In the previous section, we presented the approach to (i) understanding the differences in key attributes of HFPS respondents and the general adult population as captured in the pre-COVID-19 F2F surveys, and (ii) assessing how well individual-level weight adjustments can reduce these differences. The analysis in the previous section considers the individual-level variables used in the logit regression as part of creating the individual-level adjusted weights. This analysis is therefore a test of how well the adjustment model discussed in section 2.4 worked on a subset of its own model parameters.

In this section, we expand the analysis beyond these individual attributes from the F2F data to a practical application in using the HFPS phone survey data in a way many analysts might, which serves as a validation of our initial results. Most individual-level analyses using phone survey data face the constraint that the data are only available for the main respondent. However, we leverage the fifth HFPS round in Malawi and Nigeria, where individual-level data on labor market outcomes were collected for all adult household members and not just the main respondent. This special case allows us to create an alternative benchmark for the general adult population, which we use to understand (i) the differences in outcomes measured in the HFPS data between the HFPS respondents and the general adult population and (ii) how well the individual-level adjusted weight can overcome these differences. For this, we weight the individual-level HFPS data on select employment outcomes using the standard HFPS household sampling weights (*w1*) and assume these to be the alternative benchmark estimates for the general adult population. The HFPS household sampling weights (*w1*) are calibrated to provide representative estimates for the general household population as discussed in section 2.2 and demonstrated in a forthcoming study [15]. As such, the assumption is that the HFPS individual-level data on adult household members that are weighted by *w1* are again representative of the general adult population. We consider this assumption reasonable for the illustrative purpose of this analysis, but the caveat to consider is that the data for all adult household members is collected through the main HFPS respondent rather than from each household member directly. Collecting individual-level data through a proxy is considered second-best to self-reporting because proxy response may lead to non-sampling error [17], which may not be

mitigated through reweighting [21]. Ideally, individual-level analyses would therefore rely on self-reported data for all household members, but this may be prohibitive in the context of a phone survey. In the absence of self-reported data for all household members, we consider the proxy-reported data for all household members a reasonable alternative benchmark against which to test the outcomes for phone survey respondents.

With this setup, we compare (i) individual employment outcomes for all adults as reported by a proxy and weighted by the household sampling weight (*w1)*, the benchmark, to (ii) the same set of employment outcomes for only the sample of HFPS respondents using the household sampling weight (*w1*), to assess the differences between these two populations, to (iii) the sample of HFPS respondents using the adjusted individual (*w2*) sampling weight, to assess how well it performs in overcoming the differences. The approach to gauging graphical and statistically significant mean differences between the three competing estimates for each employment outcome is identical to the approach that is detailed in section 2.3.

## 3. Results

In the following, we first discuss how phone survey respondents differ from the general adult population and then explore how well the different weight adjustment techniques perform in allowing the data on HFPS respondents to be more representative of the general adult population.

### 3.1. Phone survey respondents versus the general adult population

Given the respondent selection protocols discussed above, it is expected that the two populations–phone survey respondents and the general adult population–differ along various dimensions. As a reminder, S1 and S2 Tables shows a set of descriptive statistics for the individual-level variables of interest for both populations in each of the four countries. Table 3 presents the results (i.e. marginal effects) from the estimation of Eq 1, i.e. the logit regression that models the likelihood of being a HFPS respondent among adults in successfully interviewed HFPS households as a function of a rich set of individual and household attributes. Several overarching results emerge.

First, household heads are most likely to be respondents. In all surveys, being the household head has the largest effect on the conditional probability of being the phone survey respondent, increasing that probability by between 31.4 percent in Nigeria and 45.7 percent in Ethiopia (with Malawi- and Uganda-specific impacts being estimated at 39.7 percent and 38.9 percent, respectively). Note that this result already accounts for phone ownership, which is one of the control variables. Being the spouse of the household head also has a large effect in all countries but Nigeria, ranging between 12.8 percent in Ethiopia and 18.3 percent in Uganda. These results are likely driven by the country-specific respondent selection protocols, which tend to favor the household head or their spouse, as discussed in section 2.1. Conditional on household headship and the remaining control variables, men are less likely to be HFPS respondents in Malawi and Uganda, and just as likely as women in being HFPS respondents in Ethiopia and Nigeria. However, men make up the majority of respondents in all four countries (Table 2). This finding is due to household heads being predominantly male combined with the strong effect headship has on being the respondent. The household head effect thus masks the gender dynamics of phone survey response.

Second, it is notable that the household head effect is similar in magnitude in Malawi as in the other three countries (Malawi: 0.397 vs Ethiopia: 0.457; Nigeria: 0.314; Uganda: 0.389), even though the Malawi survey stands out for not targeting the household head as first contact bur rather calling available phone numbers in random order. In spite of this protocol, 79 percent of respondents are household heads in Malawi, not very different from the shares of the other countries (Ethiopia: 83 percent; Nigeria: 82 percent; Uganda: 74 percent). This is due to

**Table 3. Marginal effects from logit regressions on being a HFPS respondent in round 1.**

| | Ethiopia | Malawi | Nigeria | Uganda |
|---|---|---|---|---|
| Household Size | -0.015 (.002)*** | -0.013 (.002)*** | -0.011 (.001)*** | -0.012 (.002)*** |
| Head † | 0.457 (.018)*** | 0.397 (.026)*** | 0.314 (.019)*** | 0.389 (.027)*** |
| Spouse of head † | 0.128 (.023)*** | 0.140 (.033)*** | -0.010 (.023) | 0.183 (.032)*** |
| Child of head † | 0.083 (.019)*** | -0.006 (.026) | 0.026 (.021) | 0.000 (.027) |
| Male † | -0.005 (.009) | -0.040 (.015) *** | 0.013 (.013) | -0.050 (.012)*** |
| Ages 25–49 † | 0.031 (.011)*** | 0.040 (.016) ** | 0.079 (.016)*** | 0.112 (.019)*** |
| Ages 50+ † | -0.009 (.014) | 0.038 (.019) ** | 0.060 (.018)*** | 0.094 (.020)*** |
| Married † | -0.016 (.012) | -0.021 (.019) | 0.033 (.014)** | -0.065 (.017)*** |
| Primary † | 0.030 (.010)*** | 0.005 (.015) | 0.021 (.012)* | 0.029 (.010)*** |
| Secondary † | 0.043 (.013)*** | 0.014 (.017) | 0.031 (.012)*** | 0.050 (.033) |
| Certificate † | 0.079 (.037)** | -0.002 (.016) | 0.057 (.017)*** | 0.037 (.022)* |
| Post-Secondary Degree † | 0.063 (.016)*** | 0.002 (.023) | 0.036 (.019)* | -0.003 (.020) |
| Employed for a wage/salary † | -0.007 (.010) | -0.005 (.015) | 0.039 (.013)*** | 0.007 (.012) |
| Owner of a household enterprise † | 0.026 (.011)** | 0.047 (.012)*** | 0.057 (.009)*** | 0.029 (.010)*** |
| Casual laborer † | 0.075 (.020)*** | 0.055 (.012)*** | | |
| Consumption quintile 2 † | -0.011 (.017) | -0.007 (.021) | -0.022 (.015) | -0.024 (.015) |
| Consumption quintile 3 † | -0.018 (.016) | -0.017 (.021) | -0.034 (.015)** | -0.031 (.015)** |
| Consumption quintile 4 † | -0.031 (.016)* | -0.027 (.020) | -0.042 (.016)** | -0.048 (.015)*** |
| Consumption quintile 5 † | -0.043 (.017)*** | -0.048 (.021)** | -0.041 (.017)** | -0.055 (.017)*** |
| Individual owns a mobile phone † | 0.114 (.009)*** | 0.154 (.012)*** | 0.077 (.014)*** | 0.139 (.010)*** |
| Spatial Fixed Effects | Region x Urban | District | State | Subregion |
| Number of Observations | 8535 | 4959 | 6183 | 6647 |
| Pseudo R-squared | 0.456 | 0.437 | 0.484 | 0.386 |

Note

† denotes dichotomous variables. Standard errors are reported in parentheses.

***/**/* denote statistical significance at the 1/5/10 percent level, respectively. For each country the sample is all F2F survey household members age 15 and older for the set of households that were successfully interviewed in round 1 of the phone survey.

a combination of factors. On the one hand, phone ownership is skewed towards household heads, so household heads are more likely to be called than other members in the first place.

In the Malawi sample, close to 60 percent of mobile phone owners are household heads and, in a multivariate logit regression, household heads are found to be 32 percent more likely to own a mobile phone, all other things being equal (S3 Table). On the other hand, calling available phone numbers in random order affects who is a household's first contact; but not all first contacts also ended up being the main respondent. In round 1 of the Malawi HFPS, 66 percent of main respondents were also first contacts. For the remaining 34 percent, the first contact handed the phone to a household member who then became the main respondent. One scenario is when the first contact was not a household member but a reference contact outside of the household because no one in the household owned a mobile phone (see section 2.1). This was the case for about 15 percent of households contacted for round 1 of the Malawi HFPS. Not being a member of the household, the reference contact cannot be the main respondent and so the phone was handed to a member of the household instead. In another scenario, although the first contact was a member of the household, they preferred for another member, often the household head, to be the main respondent.

Third, ownership of a mobile phone increases the probability of being the respondent substantially, ranging from 7.7 percent in Nigeria to 15.4 percent in Malawi (with Ethiopia- and

Uganda-specific impacts being estimated at 11.4 percent and 13.9 percent, respectively). This is not surprising in a phone survey context, though the effect is not as strong as the effect of household headship. Consequently, it suggests that phones are handed over from one household member to another to complete the interview.

Fourth, HFPS respondents are more educated than non-respondents in all countries except for Malawi. In Ethiopia and Nigeria, holding any of primary, secondary, post-secondary certificate, or post-secondary degrees increases probability of being a HFPS respondent vis-à-vis adults with no degree. In Uganda, there are effects specifically associated with having primary education and with having a post-secondary certificate. The effect sizes range from two to eight percent.

Fifth, being in an age category older than 15–24 increases the probability of being a phone survey respondent in all countries but Ethiopia, where individuals aged 50+ are not any more likely to be selected as HFPS respondents vis-à-vis individuals aged 15–24. The age effects are particularly pronounced in Uganda, where individuals aged 25–49 and those aged 50+ are respectively 11.2 percent and 9.4 percent more likely to be HFPS respondents compared to individuals aged 15–24.

Sixth, owning a household enterprise increases the probability of being a HFPS respondent in all countries, with the effect sizes ranging from 2.6 to 5.7 percent. The data on participation in casual labor is only available for Malawi and Ethiopia and the results show that this increases the likelihood of being a HFPS respondent by 7.5 percent in Ethiopia and 5.5 percent in Malawi. Given the high prevalence of casual labor in Malawi (estimated 38.6 percent of adults in the F2F survey), this is a relatively strong effect.

Finally, greater household wealth (proxied by household consumption quintiles) leads to a decline in the probability of being a HFPS respondent. However, differences only arise in the third quintile in Nigeria and Uganda, the fourth quintile in Ethiopia, and in the top quintile in Malawi. This suggests that wealthier households are overall less likely to respond to the phone survey, possibly due to higher opportunity cost of their time.

### 3.2 Assessing bias reduction through weight adjustments

We now turn to assessing how well the various survey weights perform at counteracting the bias associated with phone survey respondent selection. The results of the graphical analysis are shown in Figs 1–4. The effectiveness of the bias reduction is mixed and depends on the outcome of interest. Compared to the estimates obtained under the HFPS household sampling weights, the estimates based on the HFPS individual weights move closer to those for the general adult population for most individual-level outcomes of interest. However, confidence intervals widen as well. Several points stand out.

First, there are instances where the HFPS household weight (**w1**) increases the difference between the unweighted respondent data and our benchmark **wb**-weighted F2F survey sample. Notably, the incidence of headship moves further from the mean in all four countries, though the difference is easier to detect in Nigeria and Uganda. The incidence of being a spouse also shows this pattern across all countries but Uganda. Beyond headship, Ethiopia exhibits larger deviations with household weights (**w1**) than without for the estimates of the dichotomous variables identifying men and women, those in the youngest and oldest age categories and married individuals. The same is true in Malawi for the youngest and oldest age groups, Nigeria for men and women and individuals that own a household enterprise, and in Uganda for individuals in the age group 25–49, those without an educational degree, individuals that are engaged in wage employment, those that own a household enterprise, and individuals that own a mobile phone. This broad set of instances provides evidence that the HFPS household

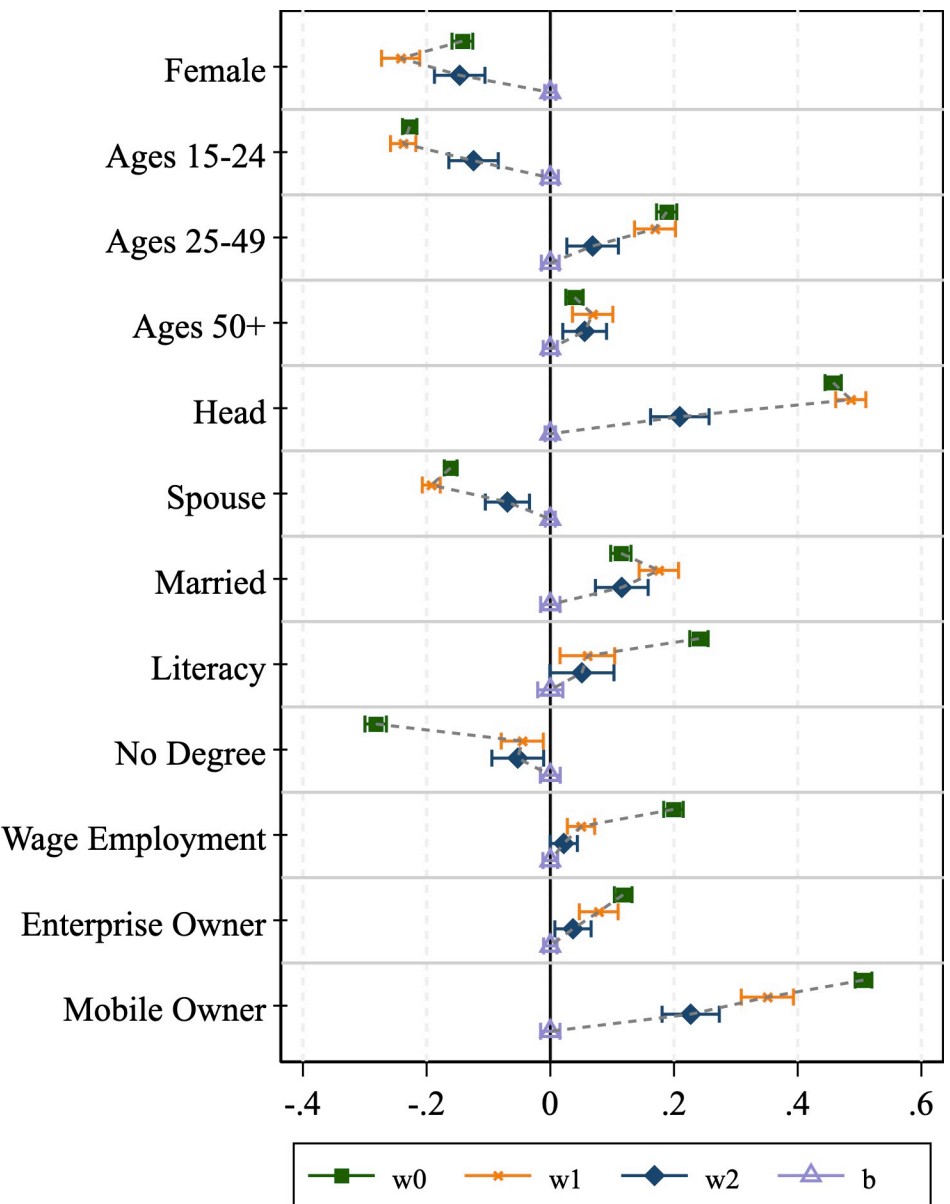

**Fig 1. Graphical inspection of bias adjustment, Ethiopia.**

weights (*w1*) do not adequately support the analysis of individual-level data on HFPS respondents in a way that is representative of the general adult population.

Second, individual weights (*w2*) substantially reduce the bias in those variables with the largest deviations from the benchmark mean. Specifically, the over-representation of household heads and mobile phone owners among phone survey respondents cannot be corrected by the HFPS household weights (*w1*) but is addressed more effectively by individual weights (*w2*). However, the individual weights only partially eliminate the difference from baseline adults and cause the confidence intervals to widen.

Lastly, there are some cases of over-correction where the individual weights move the mean estimates for the HFPS respondents beyond those that are associated with the benchmark

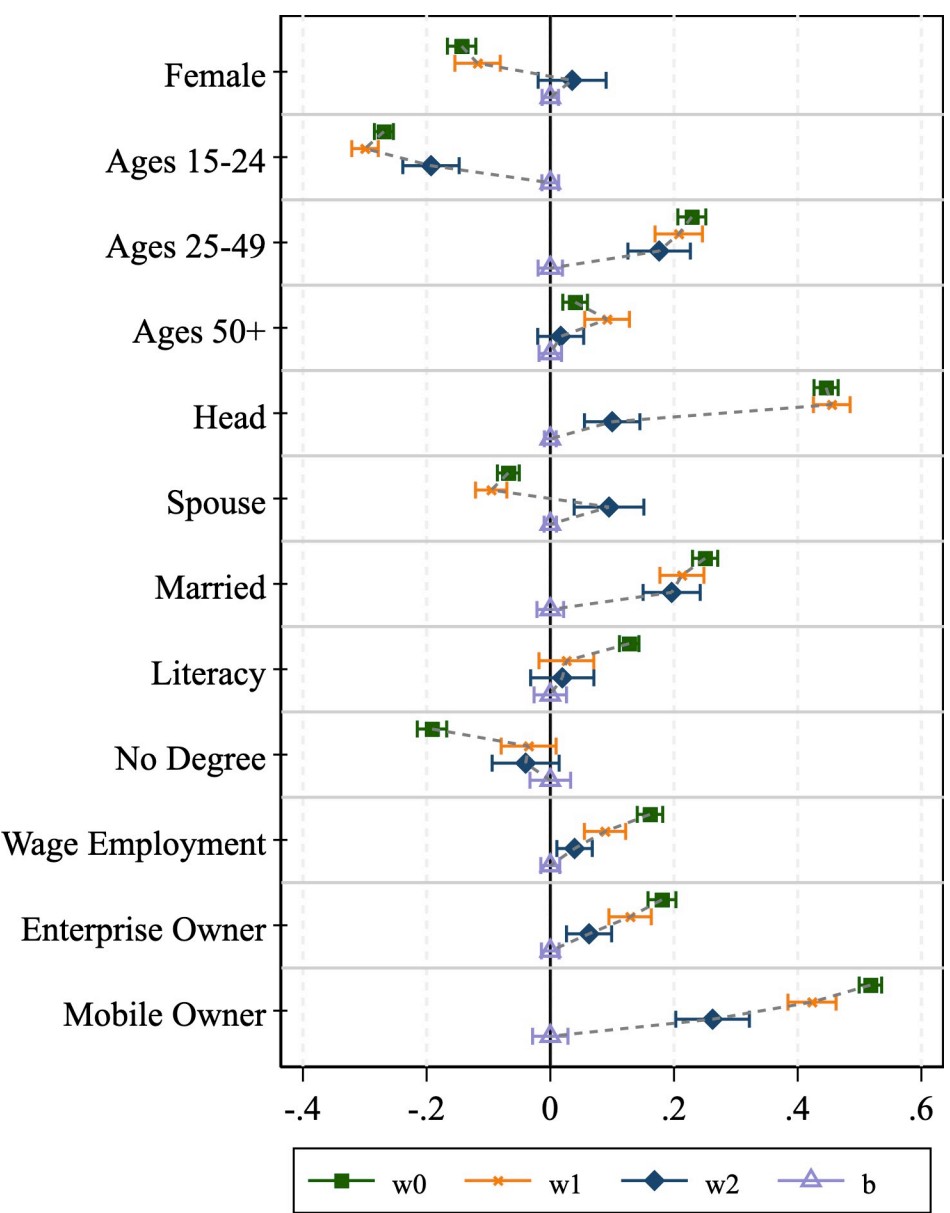

**Fig 2. Graphical inspection of bias adjustment, Malawi.**

sample of adult household members in F2F surveys. This is true particularly for the estimates of being the spouse of the household head in Malawi and being a woman in Uganda. These biases are introduced through reweighting and are not present in the unweighted data.

Tables 4 and 5 present the results from the weighted linear regressions that are detailed in section 2.3. They allow us to study whether differences between the benchmark means for the general adult population from the pre-COVID-19 F2F survey and the unweighted, household-weighted, and individual-weighted estimates for the HFPS respondents are statistically significant. The results show that the differences between the HFPS respondents and the general adult population are not fully addressed by HFPS individual weights (*w2*). However, there are a few cases where individual weights do succeed in addressing the bias. In Malawi, the individual weights can deal with over-representation of age group 50+ and under-representation of females. In all countries

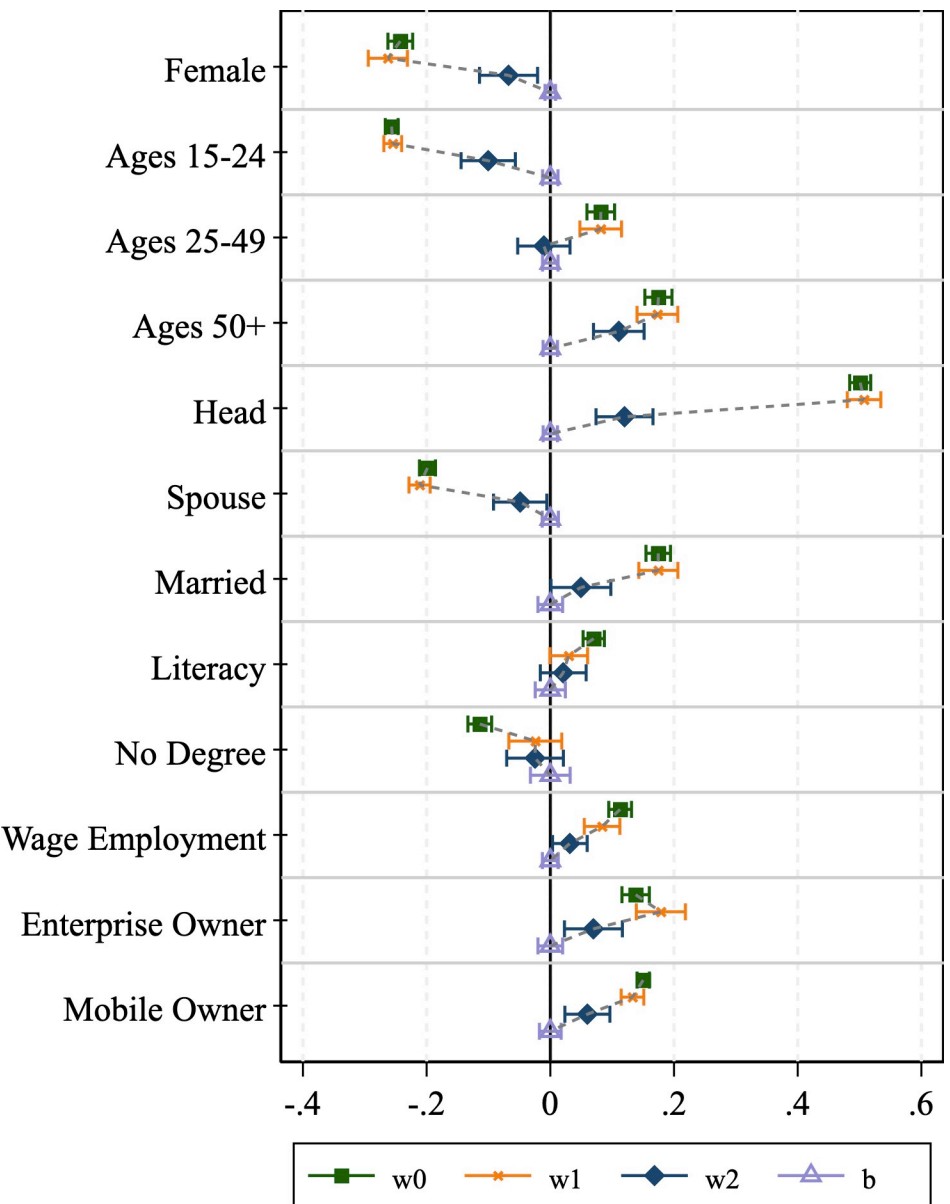

**Fig 3. Graphical inspection of bias adjustment, Nigeria.**

except for Ethiopia, under-representation of respondents without an educational degree is also mitigated. The over-arching result remains that the individual weights applied to the data on the HFPS respondents move the estimates in the right direction, but they do not successfully eliminate bias. These results hold if the sample is broken down by gender and different age groups. Gender- and age-disaggregated results are presented in S4–S7 Tables.

## 3.3. An application with individual-level employment outcomes measured in phone surveys

We now turn to the analysis of individual-level employment outcomes during COVID-19, as measured in the fifth HFPS rounds in Malawi and Nigeria. The objective is to assess the use of

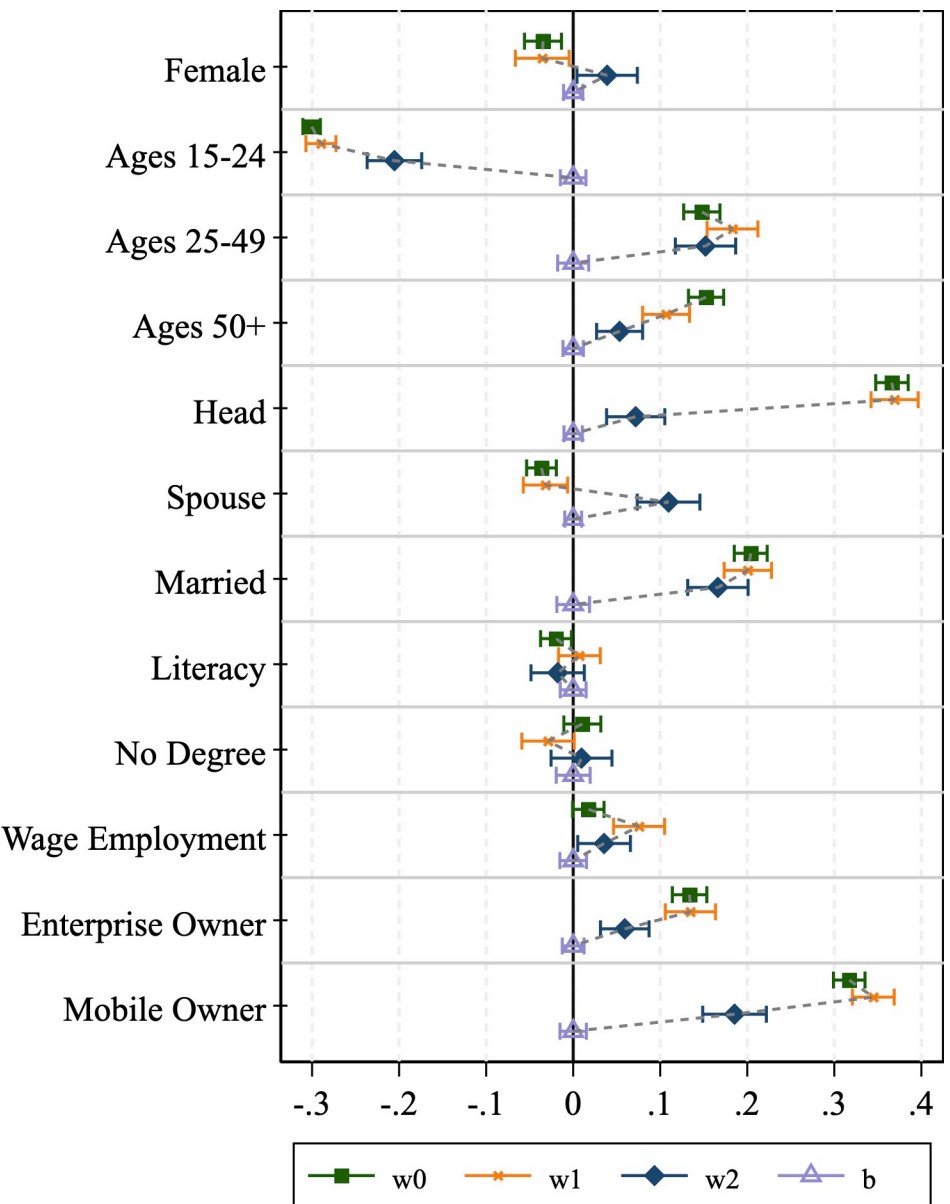

**Fig 4. Graphical inspection of bias adjustment, Uganda.**

individual-level recalibrated weights to analyze HFPS data as it would be done in many research applications. There are three dichotomous outcomes of interest that identify whether, in the past 7 days:

a. an individual worked to generate income for at least 1 hour, irrespective of type of employment (i.e. any employment),

b. an individual worked for a wage or salary (i.e. wage employment), and

c. an individual worked at a household enterprise, as an owner, manager, or a contributing laborer (i.e. self-employment).

**Table 4. Tests of mean differences between face-to-face (F2F) adults and phone respondents: Sex, age, relation to head–as measured in the F2F survey.**

| Variable | Comparison Group Sample | Comparison Group Weight | Abbrev. | Ethiopia Beta | Ethiopia p-value | Malawi Beta | Malawi p-value | Nigeria Beta | Nigeria p-value | Uganda Beta | Uganda p-value |
|---|---|---|---|---|---|---|---|---|---|---|---|
| Female | Base, All F2F Adults | F2F HH Weight | b | 0.518 | | 0.513 | | 0.515 | | 0.518 | |
| | Phone respondents | Unweighted | w0 | -0.142 | (.000)*** | -0.143 | (.000)*** | -0.243 | (.000)*** | -0.035 | (.003)*** |
| | Phone respondents | HFPS HH Weight | w1 | -0.242 | (.000)*** | -0.118 | (.000)*** | -0.263 | (.000)*** | -0.036 | (.020)** |
| | Phone respondents | HFPS Individual Weight | w2 | -0.146 | (.000)*** | 0.035 | (.206) | -0.068 | (.004)*** | 0.039 | (.028)** |
| Ages 15–24 | Base, All F2F Adults | F2F HH Weight | b | 0.356 | | 0.387 | | 0.313 | | 0.360 | |
| | Phone respondents | Unweighted | w0 | -0.227 | (.000)*** | -0.269 | (.000)*** | -0.256 | (.000)*** | -0.300 | (.000)*** |
| | Phone respondents | HFPS HH Weight | w1 | -0.238 | (.000)*** | -0.300 | (.000)*** | -0.255 | (.000)*** | -0.290 | (.000)*** |
| | Phone respondents | HFPS Individual Weight | w2 | -0.124 | (.000)*** | -0.193 | (.000)*** | -0.100 | (.000)*** | -0.205 | (.000)*** |
| Ages 25–49 | Base, All F2F Adults | F2F HH Weight | b | 0.478 | | 0.427 | | 0.469 | | 0.450 | |
| | Phone respondents | Unweighted | w0 | 0.188 | (.000)*** | 0.229 | (.000)*** | 0.082 | (.000)*** | 0.148 | (.000)*** |
| | Phone respondents | HFPS HH Weight | w1 | 0.169 | (.000)*** | 0.208 | (.000)*** | 0.082 | (.000)*** | 0.183 | (.000)*** |
| | Phone respondents | HFPS Individual Weight | w2 | 0.068 | (.001)*** | 0.176 | (.000)*** | -0.010 | (.622) | 0.152 | (.000)*** |
| Ages 50+ | Base, All F2F Adults | F2F HH Weight | b | 0.166 | | 0.186 | | 0.218 | | 0.190 | |
| | Phone respondents | Unweighted | w0 | 0.039 | (.000)*** | 0.040 | (.001)*** | 0.175 | (.000)*** | 0.153 | (.000)*** |
| | Phone respondents | HFPS HH Weight | w1 | 0.069 | (.000)*** | 0.092 | (.000)*** | 0.173 | (.000)*** | 0.107 | (.000)*** |
| | Phone respondents | HFPS Individual Weight | w2 | 0.056 | (.001)*** | 0.017 | (.342) | 0.111 | (.000)*** | 0.053 | (.000)*** |
| Head | Base, All F2F Adults | F2F HH Weight | b | 0.370 | | 0.341 | | 0.326 | | 0.374 | |
| | Phone respondents | Unweighted | w0 | 0.457 | (.000)*** | 0.446 | (.000)*** | 0.501 | (.000)*** | 0.366 | (.000)*** |
| | Phone respondents | HFPS HH Weight | w1 | 0.486 | (.000)*** | 0.455 | (.000)*** | 0.507 | (.000)*** | 0.369 | (.000)*** |
| | Phone respondents | HFPS Individual Weight | w2 | 0.209 | (.000)*** | 0.100 | (.000)*** | 0.120 | (.000)*** | 0.072 | (.000)*** |
| Spouse | Base, All F2F Adults | F2F HH Weight | b | 0.259 | | 0.232 | | 0.290 | | 0.238 | |
| | Phone respondents | Unweighted | w0 | -0.161 | (.000)*** | -0.068 | (.000)*** | -0.199 | (.000)*** | -0.036 | (.000)*** |
| | Phone respondents | HFPS HH Weight | w1 | -0.192 | (.000)*** | -0.096 | (.000)*** | -0.211 | (.000)*** | -0.032 | (.012)** |
| | Phone respondents | HFPS Individual Weight | w2 | -0.069 | (.000)*** | 0.095 | (.001)*** | -0.049 | (.034)** | 0.110 | (.000)*** |

**Note:** Base row reports the nationally representative mean among all adults in the face-to-face (F2F) survey. Rows other than the base row report the difference from the base and the p-value from a test of significance for that difference. Sample: all adults in F2F surveys, of which phone survey respondents are a sub-sample.

The pool of HFPS respondents differs slightly in round 5 vis-à-vis round 1 due to attrition. Therefore, we generate a round 5-specific HFPS individual weight, following the same steps outlined in section 2.2. Fig 5 shows the mean and confidence interval for each employment outcome of interest for:

i. all adults that were interviewed in the F2F survey and that were residing in HFPS households successfully interviewed in round 5, weighted by the round 5 HFPS household sampling weight (*w1*)–assumed to be representative of the general adult population,

ii. the main HFPS respondents interviewed in round 5, weighted by the round 5 HFPS household sampling weight (*w1*), and

iii. the main HFPS respondents interviewed in round 5, weighted by the round 5 HFPS individual sampling weight (*w2*).

We compare (i) an estimate that is assumed to be representative of the general adult population but relies on reports from a proxy, the main HFPS respondent, to (ii) a "naive" estimate applying household survey weights to the sample of HFPS respondents (based on self-reports), and (iii) estimates obtained by weighting that same sample of HFPS respondents with adjusted individual weights.

**Table 5. Tests of mean differences between face-to-face (F2F) adults and phone respondents: Marital status, education, employment–as measured in the F2F survey.**

| | Comparison Group | | | Ethiopia | | Malawi | | Nigeria | | Uganda | |
|---|---|---|---|---|---|---|---|---|---|---|---|
| *Variable* | *Sample* | *Weight* | *Abbrev.* | *Beta* | *p-value* | *Beta* | *p-value* | *Beta* | *p-value* | *Beta* | *p-value* |
| Married | Base, All F2F Adults | F2F HH Weight | b | 0.549 | | 0.508 | | 0.561 | | 0.525 | |
| | Phone respondents | Unweighted | w0 | 0.114 | (.000)*** | 0.250 | (.000)*** | 0.175 | (.000)*** | 0.204 | (.000)*** |
| | Phone respondents | HFPS HH Weight | w1 | 0.176 | (.000)*** | 0.213 | (.000)*** | 0.175 | (.000)*** | 0.201 | (.000)*** |
| | Phone respondents | HFPS Individual Weight | w2 | 0.116 | (.000)*** | 0.196 | (.000)*** | 0.050 | (.034)** | 0.166 | (.000)*** |
| Literate | Base, All F2F Adults | F2F HH Weight | b | 0.520 | | 0.747 | | 0.751 | | 0.795 | |
| | Phone respondents | Unweighted | w0 | 0.240 | (.000)*** | 0.128 | (.000)*** | 0.070 | (.000)*** | -0.020 | (.046)** |
| | Phone respondents | HFPS HH Weight | w1 | 0.060 | (.001)*** | 0.026 | (.206) | 0.030 | (.026)** | 0.007 | (.488) |
| | Phone respondents | HFPS Individual Weight | w2 | 0.051 | (.027)** | 0.019 | (.433) | 0.021 | (.253) | -0.018 | (.203) |
| No degree | Base, All F2F Adults | F2F HH Weight | b | 0.768 | | 0.676 | | 0.355 | | 0.465 | |
| | Phone respondents | Unweighted | w0 | -0.282 | (.000)*** | -0.191 | (.000)*** | -0.114 | (.000)*** | 0.011 | (.366) |
| | Phone respondents | HFPS HH Weight | w1 | -0.045 | (.003)*** | -0.035 | (.053)* | -0.024 | (.076)* | -0.029 | (.022)** |
| | Phone respondents | HFPS Individual Weight | w2 | -0.053 | (.008)*** | -0.040 | (.124) | -0.025 | (.218) | 0.010 | (.544) |
| Wage employment | Base, All F2F Adults | F2F HH Weight | b | 0.090 | | 0.089 | | 0.100 | | 0.219 | |
| | Phone respondents | Unweighted | w0 | 0.199 | (.000)*** | 0.161 | (.000)*** | 0.113 | (.000)*** | 0.017 | (.056)* |
| | Phone respondents | HFPS HH Weight | w1 | 0.050 | (.000)*** | 0.089 | (.000)*** | 0.084 | (.000)*** | 0.076 | (.000)*** |
| | Phone respondents | HFPS Individual Weight | w2 | 0.022 | (.029)** | 0.039 | (.002)*** | 0.032 | (.011)** | 0.035 | (.008)*** |
| Enterprise owner | Base, All F2F Adults | F2F HH Weight | b | 0.098 | | 0.160 | | 0.281 | | 0.185 | |
| | Phone respondents | Unweighted | w0 | 0.118 | (.000)*** | 0.181 | (.000)*** | 0.138 | (.000)*** | 0.134 | (.000)*** |
| | Phone respondents | HFPS HH Weight | w1 | 0.078 | (.000)*** | 0.129 | (.000)*** | 0.179 | (.000)*** | 0.135 | (.000)*** |
| | Phone respondents | HFPS Individual Weight | w2 | 0.037 | (.004)*** | 0.063 | (.000)*** | 0.070 | (.001)*** | 0.059 | (.000)*** |
| Mobile owner | Base, All F2F Adults | F2F HH Weight | b | 0.307 | | 0.305 | | 0.797 | | 0.445 | |
| | Phone respondents | Unweighted | w0 | 0.506 | (.000)*** | 0.518 | (.000)*** | 0.150 | (.000)*** | 0.317 | (.000)*** |
| | Phone respondents | HFPS HH Weight | w1 | 0.351 | (.000)*** | 0.423 | (.000)*** | 0.133 | (.000)*** | 0.345 | (.000)*** |
| | Phone respondents | HFPS Individual Weight | w2 | 0.227 | (.000)*** | 0.262 | (.000)*** | 0.060 | (.001)*** | 0.185 | (.000)*** |

**Note:** Base row reports the nationally representative mean among all adults in the face-to-face (F2F) survey. Rows other than the base row report the difference from the base and the p-value from a test of significance for that difference. Sample: all adults in F2F surveys, of which phone survey respondents are a sub-sample.

The mean for (i), which we take as the benchmark estimate in this portion of our analysis, is subtracted from all estimates. Following the approach in Figs 1–4, values for (ii) and (iii) thus reflect the deviation from the benchmark. Similar to the results presented in section 3.2, the HFPS individual weights succeed in moving the estimates for the HFPS respondents closer to those for the general adult population (except for the incidence of self-employment in Malawi), albeit with widened confidence intervals (Fig 5). When weighted with the HFPS household sampling weights (*w1*), i.e. the "naïve" estimate, the mean differences remain statistically significant between the estimates for the HFPS respondents and the estimates for all adults residing in HFPS households (Table 6). This result holds regardless of the country and employment outcome of interest. Once weighted with the HFPS individual weights (*w2*), the estimates of wage employment and self-employment for the HFPS respondents in Nigeria are statistically indistinguishable from the benchmark estimates. However, for the overall employment variable in Nigeria and for all three employment variables in Malawi, the mean differences remain statistically significant between the *w2*-weighted estimates for the HFPS respondents and the benchmark estimates for all adults residing in HFPS households.

The disaggregated employment results in S8 and S9 Tables are consistent with the findings presented in Table 6. In Nigeria, the individual weights remove the differences in estimates for

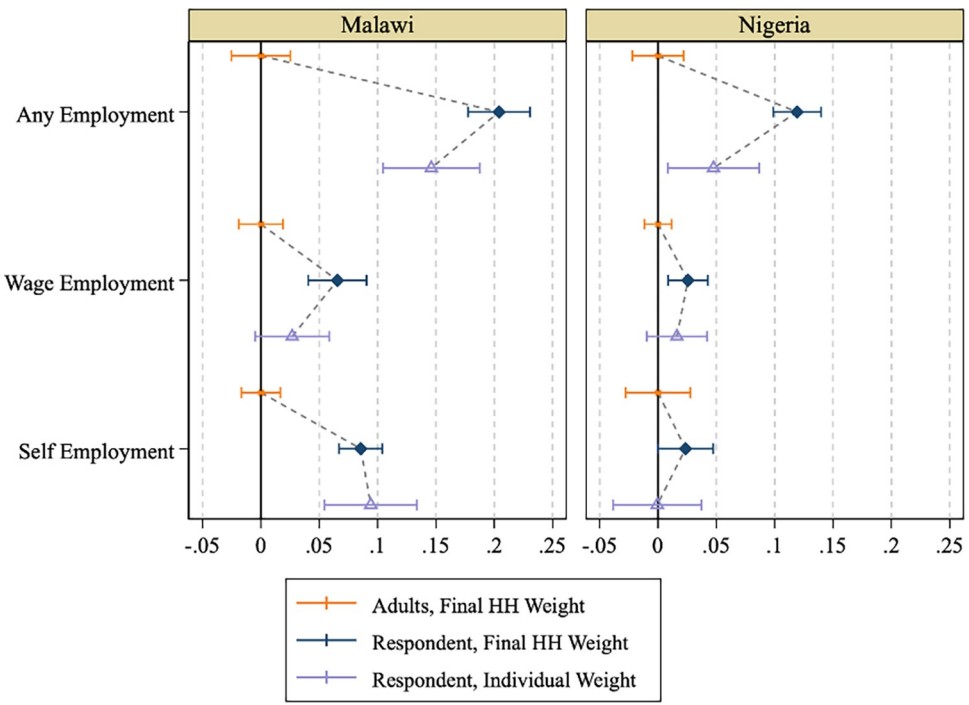

**Fig 5. HFPS round 5 employment outcomes, with HFPS household versus individual weights.**

the HFPS respondents versus the general adult population for wage employment and for self-employment, except among individuals aged 25–49 where a significant difference remains for wage employment. The differences in overall employment variable remain significant among the male sub-population, and individuals aged 25–49 in Nigeria, which are the largest sub-populations of HFPS respondents. In Malawi, the individual-weighted estimates for the HFPS respondents are statistically indistinguishable from the benchmark estimates for wage employment among males and for overall and wage employment among individuals aged 25–49. The

**Table 6. Differences in HFPS round 5 employment outcomes between phone survey respondents and adult population.**

|  | Comparison Group | | | Malawi | | Nigeria | |
|---|---|---|---|---|---|---|---|
| *Variable* | *Sample* | *Weight* | *Abbrev.* | *Beta* | *p-value* | *Beta* | *p-value* |
| Any Employment | Adults (base) | Final HH Weight | w1 | 0.612 | | 0.72 | |
|  | Respondents | Final HH Weight | w1 | 0.204 | (.000)*** | 0.119 | (.000)*** |
|  | Respondents | Individual Weight | w2 | 0.146 | (.000)*** | 0.048 | (.018)** |
| Wage Employment | Adults (base) | Final HH Weight | w1 | 0.158 | | 0.086 | |
|  | Respondents | Final HH Weight | w1 | 0.066 | (.000)*** | 0.026 | (.003)*** |
|  | Respondents | Individual Weight | w2 | 0.027 | (.099)* | 0.016 | (.220) |
| Self-Employment | Adults (base) | Final HH Weight | w1 | 0.119 | | 0.297 | |
|  | Respondents | Final HH Weight | w1 | 0.086 | (.000)*** | 0.024 | (.050)* |
|  | Respondents | Individual Weight | w2 | 0.094 | (.000)*** | -0.001 | (.975) |

Note: Base row reports the nationally representative mean among all adults present in both the F2F and phone surveys. Rows other than the base row report the difference from the base in the sample of respondents present in both F2F and phone surveys, and a p-value from a test of significance for that difference. Employment = 1 if individual spent any time in the last seven days doing specified work, 0 otherwise. All employment data are from wave 5 post-COVID survey in Malawi or wave 5 post-COVID survey in Nigeria.

HFPS household weights also mitigate bias in some subpopulations, particularly in Nigeria, but there are no cases where the individual weights do not perform at least as well. Overall, these results suggest that while individual weights can be more effective than household weights in reducing the bias in the analysis of individual level data on the main HFPS respondents, they are still insufficient to eliminate the bias in full.

## 4. Conclusion

Our analysis has confirmed that phone survey respondents in Ethiopia, Malawi, Nigeria, and Uganda are significantly different from the general adult population in a range of demographic, education, and labor market characteristics. On average, respondents are significantly more likely to be household heads or their spouses, and they tend to be older, more educated, and more likely to own a household enterprise.

We then assess how well reweighting can address these selection biases. For this, we recalibrate the HFPS household sampling weights based on propensity score adjustments derived from a cross-country comparable model of an adult individual's likelihood of being interviewed, as a function of both individual- and household-level attributes. The individual-level reweighting reduces the bias, consistently moving the estimates for the phone survey respondents closer to those for the general adult population for a range of variables. However, individual-level reweighting fails to fully overcome the biases in most cases as the differences in means remain statistically significant for most of the outcomes of interest.

An application of the individual-level recalibrated weights to the phone survey data serves as a validation of our initial results. Using individual-level phone survey data, we show that respondents' labor market outcomes during the COVID-19 pandemic differ from the adult population living in phone survey households. Here, too, individual-level reweighting is a step in the right direction but is ultimately insufficient.

Our findings have implications both for the use of existing phone survey data for individual-level analysis and for the design of future phone surveys. Phone surveys have proven critical tools to meet the urgent demand for data on the impacts of the COVID-19 pandemic in low- and middle-income countries and phone survey data have been used widely to provide insights on a broad range of issues related to the pandemic.

Across Ethiopia, Malawi, Nigeria, and Uganda alone, a total of 42 national phone survey rounds have been implemented from April 2020 to June 2021, amounting to a total of over 81,000 interviews. In the same timeframe, 28 analytical survey reports, several World Bank publications, cross-country journal publications, working papers and policy briefs were produced, with a total download count of over 21,000 [38, 41–45]. The phone surveys in the four countries we study represent a fraction of the phone surveys fielded in the context of the COVID-19 pandemic. Many national statistical offices reported implementing phone surveys [3], the World Bank supported phone surveys in over 100 countries, and many other organizations rolled out phone surveys of their own.

Our results are relevant for individual-level analyses undertaken with these data. Specifically, where phone surveys are based on a frame of phone numbers from a previous F2F survey, making full use of the available information to recalibrate weights at the individual-level is worthwhile to achieve better representativeness. The availability of information on both individuals who participate in the survey and individuals who do not is an important advantage of using phone numbers from a previous F2F survey. There are also reweighting techniques for phone surveys based on random digit dialing (RDD) [21, 29], which was used frequently for COVID-19 related phone surveys in low- and middle-income countries. However, we are not aware of any systematic attempt at assessing their effectiveness in the context at hand.

In any case, in phone surveys in which individual-level data are available only for the main respondent and respondent selection was not based on a probability sampling method, our findings suggest that achieving fully representative individual-level estimates is unlikely to be ultimately feasible and researchers should be aware of these limitations.

The rapid design and successful implementation of high-frequency phone surveys during the COVID-19 pandemic has been an unprecedented learning experience on the part of national statistical offices in low- and middle-income countries and international agencies and donor organizations that have provided financial and technical support to these operations. Phone surveys are therefore expected to be part of the post-pandemic survey landscape in low- and middle-income countries, complementing face-to-face surveys.

In view of our findings regarding the limits of representativeness of individual-level phone survey data in four African countries, survey implementers should think more critically about respondent selection protocols in future phone surveys.

A desirable option is to randomly select an adult household member to be interviewed in each household on topics that are related to individuals and personal experiences. In the context of the on-going HFPS rounds and future phone surveys that use existing household surveys as sampling frames, the interview target can be selected at random (without replacement) in each household following a household roster update. Upon the selection of the interview target, the current phone survey respondent can be asked to (i) either pass the phone to the selected individual if he or she is available, (ii) provide a phone number for the selected individual if a person-specific phone number exists, or (iii) coordinate with the selected individual to converge on a date and time for an interview using the current respondent's phone. Depending on the objective of the study, the randomly selected household member can ultimately replace or be in addition to the main phone survey respondent. Retaining the 'most knowledgeable' household member as one of the respondents may be desirable when collecting reliable household-level data is a priority.

Attempting to interview all adult members would be yet another option. Conducting several interviews per household would impinge on the comparatively low costs of phone surveys though and likely increase the scope for non-response. Given limited prior experiences with such variations in respondent selection protocols, a sensible first step would be to pilot one or several of these improved options in a random subset of households in future phone surveys to better understand the subsequent impacts on consent, non-response and attrition.

## Supporting information

**S1 Table. Descriptive statistics.** Ethiopia and Malawi. Notes: No weights are used. † denotes a dichotomous variable Respondent identifies whether the individual was a HFPS respondent– set to 1 for all individuals under the Phone Respondents column. These variables originate from the pre-COVID-19 F2F survey in each country.
(PDF)

**S2 Table. Descriptive statistics.** Nigeria and Uganda. Notes: No weights are used. † denotes a dichotomous variable. Respondent identifies whether the individual was a HFPS respondent– set to 1 for all individuals under the Phone Respondents column. These variables originate from the pre-COVID-19 F2F survey in each country.
(PDF)

**S3 Table. Marginal effects from logit regressions on mobile ownership in the sampled household baseline datasets.** Notes: † denotes dichotomous variables. Standard errors are reported in parentheses. ***/**/* denote statistical significance at the 1/5/10 percent level, respectively. For

each country the sample is all F2F survey household members age 15 and older for the set of households that were successfully interviewed in round 1 of the phone survey.
(PDF)

**S4 Table. Ethiopia: Tests of difference between face-to-face adults and phone respondents, disaggregated by sex and age group.** Notes: Base row reports the nationally representative mean among all adults in the face-to-face survey. Rows other than the base row report the difference from the base and a p-value from a test of significance for that difference.
(PDF)

**S5 Table. Malawi: Tests of difference between face-to-face adults and phone respondents, disaggregated by sex and age group.** Notes: Base row reports the nationally representative mean among all adults in the face-to-face survey. Rows other than the base row report the difference from the base and a p-value from a test of significance for that difference.
(PDF)

**S6 Table. Nigeria: Tests of difference between face-to-face adults and phone respondents, by sex and age group.** Note: Base row reports the nationally representative mean among all adults in the face-to-face survey. Rows other than the base row report the difference from the base and a p-value from a test of significance for that difference.
(PDF)

**S7 Table. Uganda: Tests of difference between face-to-face adults and phone respondents, by sex and age group.** Notes: Base row reports the nationally representative mean among all adults in the face-to-face survey. Rows other than the base row report the difference from the base and a p-value from a test of significance for that difference.
(PDF)

**S8 Table. Difference between adults and phone respondent employment outcomes, by sex.** Notes: Base row reports the HFPS-based nationally representative mean among all adults present in the face-to-face and phone surveys. Rows other than the base row report the difference from the base and a p-value from a test of significance for that difference. Employment = 1 if individual spent any time in the last seven days doing specified work, 0 otherwise. All data are from the fifth round of the HFPS in Malawi and Nigeria.
(PDF)

**S9 Table. Difference between adults and phone respondent employment outcomes, by age group.** Notes: Base row reports the HFPS-based nationally representative mean among all adults present in the face-to-face and phone surveys. Rows other than the base row report the difference from the base and a p-value from a test of significance for that difference. Employment = 1 if individual spent any time in the last seven days doing specified work, 0 otherwise. All data are from the fifth round of the HFPS in Malawi and Nigeria.
(PDF)

## Acknowledgments

The authors would like to thank Kathleen Beegle, Calogero Carletto, Isabela Coelho, Kristen Himelein, and Yannick Markhof for their comments on the earlier versions of this paper. We also thank the individuals involved in the design, implementation and dissemination of high-frequency phone surveys on COVID-19, specifically the World Bank LSMS team, and the phone survey managers and interviewers at the Malawi National Statistical Office, the Nigeria Bureau of Statistics, the Uganda Bureau of Statistics and Laterite Ethiopia.

## Author Contributions

**Conceptualization:** Joshua Brubaker, Talip Kilic, Philip Wollburg.

**Data curation:** Joshua Brubaker.

**Formal analysis:** Joshua Brubaker, Talip Kilic, Philip Wollburg.

**Methodology:** Joshua Brubaker, Talip Kilic, Philip Wollburg.

**Visualization:** Joshua Brubaker, Talip Kilic, Philip Wollburg.

**Writing – original draft:** Joshua Brubaker, Talip Kilic, Philip Wollburg.

**Writing – review & editing:** Joshua Brubaker, Talip Kilic, Philip Wollburg.

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
