## [Decision Letter · Decision Letter 0]

21 Jul 2021

PONE-D-21-17113

Representativeness of individual-level data in COVID-19 phone surveys: Findings from Sub-Saharan Africa

PLOS ONE

Dear Dr. Wollburg,

Let me apologize first for taking much more time than expected. All reviewers kept requesting for extensions of deadlines and in the current situation it is hard for editors to not grant these. Even at this point, one review is still outstanding, but I decided we should be able to proceed with the two I received at this point.

From the assessments of the reviewers, you will see that both reviewers feel this is important research, but also have some important questions on the methodology used (why deciles?) and suggestions on how to make the paper stronger overall. Therefore, we invite you to submit a revised version of the manuscript that addresses the points raised during the review process.

We look forward to receiving your revised manuscript.

Kind regards,

Bjorn Van Campenhout, Ph.D.

Academic Editor

PLOS ONE

Reviewers' comments:

Reviewer #1: During the COVID-19 pandemic some of the LSMS-ISA household surveys were used as sampling bases for new phone surveys. This paper assesses the reliability of these phone surveys to provide individual-level statistics. One of the main constraints to doing this is that 74% to 83% of the phone calls were made to household heads, so the question arises how you can use this information to say something about the population as a whole. The public use data already include household-level weights that correct for non-response in the phone survey. This paper shows that those weights are not adequate to get accurate individual level representation and shows that a reweighting procedure using individual weights provides some improvement.

My main comments are about (i) framing of the paper and (ii) calculation of the weights and (iii) the tests.

(i) Clearly the main problem the authors have to deal with is that phone numbers were mainly collected from household heads and even where they were not, the interviewers ended up speaking to the heads. It is obviously not going to be easy to say something about the general population if you have such a specific sample of household heads. So the problem is really: with four fifths of the sample household heads, how can we say something about individual responses (on, say, knowledge of COVID-19) of everyone else. Presumably this is going to come about by giving lots of weight to those who are not heads and don’t have the typical head characteristics. There is no methodological innovation in this paper as such reweighing is quite standard, but because this is an important and widely used data source there is, in my opinion, merit in knowing whether we can use LSMS-ISA data for individual level analysis.

The problem is also quite specific to this survey constellation in LSMS-ISA. If the survey designers had known that a phone survey would follow, they very likely would not have set it up this way. So the question is one of how much reweighing can help in this specific survey.

These important points to highlight when framing the paper.

(ii) There is something odd about the two-step procedure you use. You estimate the probability of each individual to end up being interviewed in equation 1 and use the inverse (of the decile-average) to reweigh. But instead of applying that weight directly you multiply it with the household weight (in equation 3). I don’t understand the logic of doing this. Why not just use the inverse of the probability that each individual ends up in the sample as sampling weight? That is the tried and tested technique used widely to correct for such problems. If you do want to argue in favor of the two-step procedure then you should take account of your nested procedure when estimating equation 1.

(iii) There is something circular about your tests: you use a set X of observed characteristics of individuals to reweigh the observations. Then you see whether after reweighing this same set of characteristics X better reflects the population average. That is setting yourself mechanically up for success. And I think that if you drop the two-step procedure and use the single step procedure you will get a pretty good match. This is something to acknowledge and address in the paper.

Four specific questions:

• Could you explain why you use deciles of the predicted probabilities rather than the probabilities themselves?

• Section 2.6 is not very clearly written; I only understood what you were trying to do after reading the results section, which defeats the purpose of having a section explaining the empirical set-up. I suggest a careful rewrite.

• What do you mean that the winsorized weight are post-stratified (p19). Could you explain this part better?

• Why is winsorization necessary? You have already taken deciles for you weights so presumably there are no outliers. Could you provide some specific numbers to show what is going on in the extremes?

Reviewer #2: This article analyzes whether unweighted phone survey data are subject to selection bias, and whether the use of household- and individual-level weights correct for such bias. The paper uses household rosters obtained from pre-COVID face-to-face household surveys as nationally representative data, and as a benchmark for what the survey population of phone surveys administered during the pandemic should look like. Obviously, as the phone survey protocols targeted the household head, the unweighted phone survey sample does not resemble the nationally representative adult population. Interestingly, the use of household-level weights does not help address this bias, and in many cases, makes the phone survey sample even less representative of the adult population. Although individual-level weights perform better, this method is not sufficient as a bias correction technique, as differences between the nationally representative adult sample and the phone survey sample remain significant for several variables.

The analyses performed in this paper are interesting and worthwhile publishing, but I believe that the paper needs to be rewritten. First and foremost, I find the question of whether the phone survey data are representative of the adult population not very novel or interesting and would suggest taking that result for granted or as an obvious outcome of the survey protocols, which were not designed to get a representative sample of individuals. The protocols targeted households, and were mostly blind towards selection of respondents within the household.

The real added value of this paper is therefore the analyses of the bias correction methods: when we do our phone surveys with just one member of the household, usually the household head, can we use weighting methods to correct for a potential selection bias? The paper finds that this is not the case, and this has important implications for respondent selection protocols, if one is interested in individual- instead of household-level data. This could be focused on much more throughout the abstract, introduction, and conclusion.

For instance, it is not salient from the abstract that the focus is on selection bias from an individual rather than household point of view, and as a result, the last sentence of the abstract comes as a surprise. The introduction could be shorter and more focused on the issue at hand - that most phone surveys are done with just one person in the household, typically the household head, and thereby not representative of the adult population. The main question, then, is whether bias correction methods can make the data more representative, or whether we should adjust the methods through which we select respondents within a household. To make this salient, it would help if the paper introduced in the last paragraph on page 8 (using the same data) is introduced much earlier in the introduction.

Second, and related to the first comment, the data from Malawi appear underutilized in better understanding the drivers of the selection bias. In all other countries, the protocol was to first call the household head; but in Malawi, phone numbers were called in random order. Despite that, Malawi still has significant selection bias, and it would be good to know whether this is because household members are passing the phone to the household head, or whether certain types of household members are less likely to pick up their phone, and replaced by the next number on the list.

Third, the weighting is an important aspect of the paper, but the discussion of how the weights are constructed is difficult to follow and although I understand that the authors do not want to provide the technical details here, as they are presented in a different paper, the presentation could benefit from more intuition. For instance, why are deciles being created? What is their use (and why deciles as opposed to for instance quintiles or quartiles)? And why is post-stratification applied to the weights?

Fourth, I am not convinced by the added value of the employment outcomes. The supposedly representative data are reported by proxies, except for the data that the respondents provide for themselves. If we find differences between the individual-weighted and benchmark data, is that because the weighting does not work, or because the benchmark data are biased? The authors discuss this in the conclusion, but it raises questions around the added value of this comparison; unless the authors could show that findings are the same regardless of whether we study variables on which there should be less asymmetric information and more accurate reporting by proxies, versus variables where we would expect more error.

Finally, with the large set of variables that the authors look into, times the 4 countries for which the analyses are replicated, there are a lot of numbers to review in order to draw conclusions. Although I see why the weights are estimated at the country level, in presenting the results on bias correction, the regressions could be presented at the aggregate level, combining all countries in one regression, and the country-specific analyses could be presented in an appendix. Alternatively, would there be scope for an aggregation over the different variables, in order to reduce the number of coefficients that need to be interpreted and aggregated by eyeballing before drawing conclusions?

Other comments:

- The first part of the introduction appears negligent of a large body of literature on phone surveys and survey design; but in fact this literature is mentioned towards the end of the introduction. In this case, I would consider integrating the existing literature in the first part of the introduction, and presenting the findings along with their implication as the key contribution, to shorten the introduction and appear less negligent of the work that has already been done in this field.

- The figures were not visible in the manuscript itself, only as the appendices of the submission.

- A bit more explanation on why certain things vary across countries would be useful. For instance, why are the fifth round data for two countries excluded? Did these not include the employment outcomes? But how does that rhyme with all surveys being standardized across the four countries? And why were the selection protocols different in the four countries?

- At times, the paper reads a bit like a promotional brochure for the World Bank (for instance the introduction claims that the World Bank is the prime institute doing and learning around phone surveys, followed by 3-4 other institutions, among others). The fact that the anonymized surveys are published online a few weeks after completion could be a footnote in the methods section and does not require an entire paragraph. The paragraph in the conclusion stating how widely these data are used is irrelevant unless it is used to illustrate that people are using the data for individual-level analyses and therefore using the wrong household-level weights.

- Bottom of page 19 (last sentence), note that there is a small typo: households instead of household.

- IRB: Clarify that the data to which the authors had access were anonymized and that the merging of phone survey and face-to-face data was based on an anonymized household ID. Even if for data collection no ethics approval was needed, for a researcher to work with data that contain personal identifiers, and merge data from different sources, IRB should be obtained.

Overall, though, the paper presents an extremely useful analysis that needs to be published, since we should be more aware that our household-level weight adjustments often do more harm than good if using individual-level data, and that survey respondent selection protocols need to be adjusted when the objective is individual-level analysis.

---

## [Author Response · Author response to Decision Letter 0]

4 Sep 2021

Responses to Comments from Reviewer #1

Comment: During the COVID-19 pandemic some of the LSMS-ISA household surveys were used as sampling bases for new phone surveys. This paper assesses the reliability of these phone surveys to provide individual-level statistics. One of the main constraints to doing this is that 74% to 83% of the phone calls were made to household heads, so the question arises how you can use this information to say something about the population as a whole. The public use data already include household-level weights that correct for non-response in the phone survey. This paper shows that those weights are not adequate to get accurate individual level representation and shows that a reweighting procedure using individual weights provides some improvement.

My main comments are about (i) framing of the paper and (ii) calculation of the weights and (iii) the tests.

(i) Clearly the main problem the authors have to deal with is that phone numbers were mainly collected from household heads and even where they were not, the interviewers ended up speaking to the heads. It is obviously not going to be easy to say something about the general population if you have such a specific sample of household heads. So, the problem is really: with four fifths of the sample household heads, how can we say something about individual responses (on, say, knowledge of COVID-19) of everyone else. Presumably this is going to come about by giving lots of weight to those who are not heads and don’t have the typical head characteristics. There is no methodological innovation in this paper as such reweighing is quite standard, but because this is an important and widely used data source there is, in my opinion, merit in knowing whether we can use LSMS-ISA data for individual level analysis.

The problem is also quite specific to this survey constellation in LSMS-ISA. If the survey designers had known that a phone survey would follow, they very likely would not have set it up this way. So, the question is one of how much reweighing can help in this specific survey.

These important points to highlight when framing the paper.

Response: Thank you for these thoughtful comments. We agree with the fact that the issue we discuss is specific to a phone survey setup where the sampling frame is based on phone numbers collected from a previous survey and respondent selection targeted the most knowledgeable adult, as it was done in the case of the national phone surveys that have been supported by the Living Standards Measurement Study.

However, we contend that this kind of setup is relevant not only for our phone surveys of interest but for many phone surveys conducted in LMICs. In a research synthesis commissioned by IPA, Henderson and Rosenbaum, 2020 describe the most common ways phone survey sampling is conducted in LMICs. These are: (i) using an existing list of phone numbers from a previous survey or program ( the LSMS-ISA-supported pre-COVID-19 face-to-face surveys in our case), (ii) using a list of phone numbers provided by e.g. mobile network operators, (iii) random digit dialing (RDD). In the same paper, which focuses on gathering the available evidence on phone surveys in LMICs available from before the pandemic, an inventory of 21 surveys using a similar setup to the ones we discuss in our paper is presented (vs. 9 surveys using RDD vs. 3 using mobile network operator lists). In addition, in the surge of phone surveys that the COVID-19 pandemic precipitated, the practice of using phone numbers from pre-COVID-19 face-to-face surveys was quite common. Some examples of this are the IPA-led Cote d’Ivoire Recovr Survey as well as World Bank-supported High Frequency Phone Surveys in Cambodia, Chad, Mali, Burkina Faso, Kenya, Sao Tome and Principe, and Zambia. In the latter set of phone surveys supported by the World Bank, the phone survey respondent was also NOT selected randomly among the eligible adult household members (as in the case of the phone surveys that we analyze and that provide a motivation for our work).

Given the relative success of phone surveys during the COVID-19 pandemic, phone surveys are expected to play a bigger role in survey data collection in LMICs going forward. So, we view our paper as also informing these future efforts on how to collect more representative data at the individual level, and we now reflect this reasoning in the Introduction and Conclusion sections

Comment: (ii) There is something odd about the two-step procedure you use. You estimate the probability of each individual to end up being interviewed in equation 1 and use the inverse (of the decile-average) to reweigh. But instead of applying that weight directly you multiply it with the household weight (in equation 3). I don’t understand the logic of doing this. Why not just use the inverse of the probability that each individual ends up in the sample as sampling weight? That is the tried and tested technique used widely to correct for such problems. If you do want to argue in favor of the two-step procedure then you should take account of your nested procedure when estimating equation 1.

Response: Thank you for this comment. In what follows, we explain our reasoning regarding why we do the sampling weight calibration in the way that we do (and hence, why we prefer to stick to it).

First, we do not have a “nested” model per se, when we are following the “two-step” procedure. That is, we start out with the phone survey household sampling weights as given and as disseminated in the public use datasets. These weights had already been subject to (a) adjustments to counteract selection bias at the household-level and (b) post-stratification to match estimated population totals. We then augment these weights for the phone survey sample with an additional adjustment based on a cross-country comparable propensity score model that we estimate anew and that provides us with estimated probabilities of being interviewed among adult household members within the phone survey sample.

Second, our objective is to carry out a calibration exercise that an average data user can consider doing with the public use datasets. This was not clearly stated in the original manuscript and the revision rectifies this shortcoming. 

Specifically, the phone survey household sampling weights in the public use datasets readily take into account the differential selection into the HFPS sample at the household-level. What is not so obvious is that this process actually starts out by using information that is actually NOT publicly available, specifically the information on whether a phone number is available for at least one household member or a reference individual. 

The availability of this information determines the initial HFPS interview targets out of the total pre-COVID-19 survey sample. In turn, the first sampling weight adjustment that is carried out at the household-level by the data producers is to multiply the pre-COVID-19 household sampling weights for the initial HFPS interview targets with the inverse of the probability of selection into the phone survey. The probability for a given household is defined as the total number of households that was attempted to be called in household’s region divided by the total number of pre-COVID-19 survey households in that region. The first sampling weight adjustment is then augmented with an additional adjustment in order to account for non-response among the households targeted for phone interviews.

As an average data user, if you seek to carry out the single-step correction suggested by the reviewer, you still would need to apply an initial adjustment to the pre-COVID-19 sampling weight associated with each adult living in a household selected for phone interviews – along the lines of the “first sampling weight adjustment” described in the previous paragraph. You would of course not be able to do this since the information is not publicly available – that is, you do not know which adults were living in pre-COVID-19 survey households that were attempted to be called by the phone survey. The data users only know the successfully interviewed household sample and individual household members within, and they can link them (through the unique identifiers) with their pre-COVID-19 survey data.

In the revised section 2.4, we now note the following: “In what follows, we detail an approach that can be followed by any potential data user, leveraging solely the publicly available data on successfully interviewed HFPS households and their adult household members as captured in the pre-COVID-19 F2F survey and the HFPS.”

Comment: (iii) There is something circular about your tests: you use a set X of observed characteristics of individuals to reweigh the observations. Then you see whether after reweighing this same set of characteristics X better reflects the population average. That is setting yourself mechanically up for success. And I think that if you drop the two-step procedure and use the single step procedure you will get a pretty good match. This is something to acknowledge and address in the paper.

Response: Thank you for this comment. We have now made sure to acknowledge, in sections 2.5 and 2.6, that we are testing the set of individual-level variables that we also use in the reweighting model (alongside household-level variables).We contend that our analysis shows that attempting to adjust this many variables at once is quite complex and may not necessarily lead to a completely successful reweighting outcome. 

In any case, we agree with your point on testing the same set of characteristics used in the reweighting model. At the same time, the tests we perform in sections 2.6 and 3.3 with phone survey-based employment outcomes present an application of the individual-level recalibrated weights on variables that were not used in the reweighting model. These tests can be considered as a validation exercise that also mimics how analysts would use the phone survey data in their research. Specifically, in round 5 of Malawi and Nigeria we have data for all adult household members, as opposed having data only for the main respondent. This allows us to show (acknowledging some scope for proxy respondent bias)how well reweighted estimates for the main respondent align with the estimates based on all adult household members and the existing phone survey household sampling weights (i.e. without additional calibration). The original manuscript did not do a very good job at conveying this idea clearly – as you also pointed out in another comment. We have revised sections 2.6 and 3.3 and feel they are now much clearer and add a validation angle to our analysis. 

Comment: Four specific questions:

• Could you explain why you use deciles of the predicted probabilities rather than the probabilities themselves?

Response: Thank you. In using deciles of predicted probabilities, as well as trimming/winsorizing and post-stratifying, we are following the advice and best practices laid out in the sampling literature relevant to these kinds of surveys, specifically Himelein, 2014; Himelein et al., 2020; Little et al., 1997; and Rosenbaum and Rubin, 1984. On the use of deciles: the original idea belongs to Rosenbaum and Rubin (1984) and is picked up by Himelein (2014), which has provided the basis for computing longitudinal sampling weights for the surveys that have been supported by the LSMS-Integrated Surveys on Agriculture Initiative since 2008 – including for the pre-COVID-19 household surveys that served as the sampling frames for the phone surveys. The basic idea is to balance treatment (respondent) and control (non-respondents) observations on a large set of observable characteristics. Dividing observations into deciles of predicted probabilities rather than the predicted probabilities themselves, creates subgroups with equal predicted probability that contain both treated and control observations. This would not be the case with the ‘raw’ predicted probability variable since it is based on so many different variables that each observation would receive its own value. Rosenstein and Rubin (1984) discuss this issue on p. 517 of their article. We have added a sentence summarizing this reasoning briefly.

Comment: Section 2.6 is not very clearly written; I only understood what you were trying to do after reading the results section, which defeats the purpose of having a section explaining the empirical set-up. I suggest a careful rewrite.

Response: Thank you for this comment. We have carefully revised section 2.6 (as well as section 3.3, the corresponding results section). We feel this has improved its clarity and purpose. We reframed section 2.6 to make clear that it is a type of validation of our initial results, an application of the individual-level recalibrated weights to an analysis of labor market outcomes using the phone survey data in a way that researchers may use the phone survey data. The section then seeks to assess how much of a difference it makes to use the recalibrated weights and how successful they are at overcoming differences owed to the selection of respondents.

Comment: What do you mean that the winsorized weight are post-stratified (p19). Could you explain this part better?

Response: Thank you. This means that the raw weights w_{i,\\ af} are winsorized and then, in the next step, the winsorized weights are weights are post-stratified. We winsorize to deal with extreme outliers, which in turn reduces standard errors and makes estimates more efficient. We post-stratify to ensure the weights sum up to known population totals and reduce overall standard errors. These steps are anchored in the published literature on the topic (Himelein, 2014) as well as the World Bank sampling design guidance document for high-frequency phone surveys on COVID-19 (Himelein, Kastelic et al. 2020). We have clarified this in the manuscript.

Comment: Why is winsorization necessary? You have already taken deciles for you weights so presumably there are no outliers. Could you provide some specific numbers to show what is going on in the extremes?

Response: Thank you for this comment. Winsorization aims to deal with extreme outliers and therefore reduces standard errors and increases the precision of the estimates. Reducing standard errors is important in this context because reweighting always increases standard errors, and especially so with individual-level reweighting where we are trying to adjust for a quite a large distortion (e.g. a majority of respondents are household heads). We perform winsorization on the variable \\ w_{i,\\ af}={af}_{D=d}\\ast w1, which is the household level weight w1 multiplied by the adjustment factor, af. The latter is deciles of the predicted probability as discussed above. However, after multiplication with w1, outliers are well possible, and we therefore winsorize. We had also performed a number of tests with trimmed vs untrimmed weights, and while our main conclusions were qualitatively unaltered, we follow the best practices outlined in the literature. For your information, we provide a set of summary statistics of the trimmed and untrimmed w2 weight variables.

 Ethiopia Malawi Nigeria Uganda

 pre-winsorize winsorized pre-winsorize winsorized pre-winsorize winsorized pre-winsorize winsorized 

mean 15,797 12,647 6,079 4,995 42,988 314,648 10,411 8,976

min 62 134 44 69 1,010 1,393 766 1,007

p1 123 134 61 69 1,179 2,835 915 1,007

p50 3,232 3,232 1,728 1,728 13,362 354,348 3,722 3,722

p99 275,707 133,899 115,144 54,180 513,444 354,348 118,152 59,407

max 977,770 133,899 208,113 54,180 1,682,558 354,348 243,572 59,407

Responses to Comments from Reviewer #2

Comment: This article analyzes whether unweighted phone survey data are subject to selection bias, and whether the use of household- and individual-level weights correct for such bias. The paper uses household rosters obtained from pre-COVID face-to-face household surveys as nationally representative data, and as a benchmark for what the survey population of phone surveys administered during the pandemic should look like. Obviously, as the phone survey protocols targeted the household head, the unweighted phone survey sample does not resemble the nationally representative adult population. Interestingly, the use of household-level weights does not help address this bias, and in many cases, makes the phone survey sample even less representative of the adult population. Although individual-level weights perform better, this method is not sufficient as a bias correction technique, as differences between the nationally representative adult sample and the phone survey sample remain significant for several variables.

The analyses performed in this paper are interesting and worthwhile publishing, but I believe that the paper needs to be rewritten. First and foremost, I find the question of whether the phone survey data are representative of the adult population not very novel or interesting and would suggest taking that result for granted or as an obvious outcome of the survey protocols, which were not designed to get a representative sample of individuals. The protocols targeted households and were mostly blind towards selection of respondents within the household.

The real added value of this paper is therefore the analyses of the bias correction methods: when we do our phone surveys with just one member of the household, usually the household head, can we use weighting methods to correct for a potential selection bias? The paper finds that this is not the case, and this has important implications for respondent selection protocols, if one is interested in individual- instead of household-level data. This could be focused on much more throughout the abstract, introduction, and conclusion.

For instance, it is not salient from the abstract that the focus is on selection bias from an individual rather than household point of view, and as a result, the last sentence of the abstract comes as a surprise. The introduction could be shorter and more focused on the issue at hand - that most phone surveys are done with just one person in the household, typically the household head, and thereby not representative of the adult population. The main question, then, is whether bias correction methods can make the data more representative, or whether we should adjust the methods through which we select respondents within a household. To make this salient, it would help if the paper introduced in the last paragraph on page 8 (using the same data) is introduced much earlier in the introduction.

Response: Thank you for these thoughtful comments. We agree that the analyses of bias correction methods are the most important contribution of the paper. The differences between the sample of respondents and the general adult population provide the backdrop for these analyses. We have edited (and shortened) the abstract, introduction, and conclusion to improve the framing of the paper in line with your feedback. Specifically, in the introduction, we make salient the focus on individual-level data and we focus more directly on the role of bias correction. In the conclusion, we now discuss the implications of our findings for using existing data for individual-level analyses and how respondent selection may be altered going forward to improve individual-level data, if that is compatible with the objectives of the survey.

Comment: Second, and related to the first comment, the data from Malawi appear underutilized in better understanding the drivers of the selection bias. In all other countries, the protocol was to first call the household head; but in Malawi, phone numbers were called in random order. Despite that, Malawi still has significant selection bias, and it would be good to know whether this is because household members are passing the phone to the household head, or whether certain types of household members are less likely to pick up their phone, and replaced by the next number on the list.

Response: Thank you for raising this point. We agree the Malawi data can be utilized to provide additional insights. We therefore addressed this concern by adding a thorough discussion of the Malawi results in section “3.1. Phone survey respondents versus the general adult population”. For this, we gathered additional information from the country team to better understand the steps taken between first contact with the household and the first completed interview with the household. Second, we added in the Appendix a table with regressions detailing the determinants of mobile ownership similar to Table 3. Based on this, we now write in section 3.3:

“[…] it is notable that the household head effect is similar in magnitude in Malawi as in the other three countries (Malawi: 0.397 vs Ethiopia: 0.457; Nigeria: 0.314; Uganda: 0.389), even though the Malawi survey stands out for not targeting the household head as first contact bur rather calling available phone numbers in random order. In spite of this protocol, 79 percent of respondents are household heads in Malawi, not very different from the shares of the other countries (Ethiopia: 83 percent; Nigeria: 82 percent; Uganda: 74 percent). This is due to a combination of factors. On the one hand, phone ownership is skewed towards household heads, so household heads are more likely to be called than other members in the first place. In the Malawi sample, close to 60 percent of mobile phone owners are household heads and, in a multivariate logit regression, household heads are found to be 32 percent more likely to own a mobile phone, all other things being equal (S 3 Table). On the other hand, calling available phone numbers in random order affects who is a household’s first contact; but not all first contacts also ended up being the main respondent. In round 1 of the Malawi HFPS, 66 percent of main respondents were also first contacts. For the remaining 34 percent, the first contact handed the phone to a household member who then became the main respondent. One scenario is when the first contact was not a household member but a reference contact outside of the household because no one in the household owned a mobile phone (see section 2.1). This was the case for about 15 percent of households contacted for round 1 of the Malawi HFPS. Not being a member of the household, the reference contact cannot be the main respondent and so the phone was handed to a member of the household instead. In another scenario, although the first contact was a member of the household, they preferred for another member, often the household head, to be the main respondent.”

Comment: Third, the weighting is an important aspect of the paper, but the discussion of how the weights are constructed is difficult to follow and although I understand that the authors do not want to provide the technical details here, as they are presented in a different paper, the presentation could benefit from more intuition. For instance, why are deciles being created? What is their use (and why deciles as opposed to for instance quintiles or quartiles)? And why is post-stratification applied to the weights?

Response: Thank you for this comment. We have tried to add some more intuitive reasoning especially to the questions of winsorization and post-stratification. Winsorization is done to deal with extreme outliers, which reduces standard errors and makes estimates more efficient. Post-stratification serves the purpose of (i) ensuring the weights sum up to known population totals and (ii) further reduce overall standard errors, which is important for this paper since reweighting increases standard errors, especially in this application. As noted in response to similar comments from Reviewer #1, our empirical approach is anchored in the recommendations in the peer-reviewed literature. 

As for the use of deciles, please see our response above to a similar comment from Reviewer #1. We added a sentence that summarizes the reasoning, though for a more detailed account, readers should consult the relevant cited literature. 

Comment: Fourth, I am not convinced by the added value of the employment outcomes. The supposedly representative data are reported by proxies, except for the data that the respondents provide for themselves. If we find differences between the individual-weighted and benchmark data, is that because the weighting does not work, or because the benchmark data are biased? The authors discuss this in the conclusion, but it raises questions around the added value of this comparison; unless the authors could show that findings are the same regardless of whether we study variables on which there should be less asymmetric information and more accurate reporting by proxies, versus variables where we would expect more error.

Response: Thank you for this comment. We have revisited the phone survey employment outcomes sections (in 2.6 and 3.3) to decide how these results may best be used to add value to the overall analysis – also in light of a set of comments from reviewer #1. The write up of these sections lacked in clarity and direction and so we have decided to reframe it slightly: We see the value added of this part of the analysis in the fact that we are using actual phone survey data to validate the use of individual-level weights in a setup that mimics the kind of analysis that researchers and analysts would actually use the phone survey data for. In the specific case of round 5 in Malawi and Nigeria, we have individual-level data for all adult household members, but in the overwhelming majority of situations, analysts and researchers will have individual-level data only for the main respondent – that is at the heart of the motivation for this paper. As part of the unique set-up in round 5 in Malawi and Nigeria, we can create an alternative benchmark of the general adult population and assess how the differences between the respondents and the general adult population likely affect individual-level analyses with phone survey data when the data are only available for main respondents. We also assess how well the individual-level weights fare at reducing these differences. With this framing, we do now think this part adds value to the overall analysis.

Having said that potential proxy response bias is an important caveat that we acknowledge in the section. The paper cites Kilic et al. (2020) who, for Malawi, provides suggestive evidence of underreporting of wage- and self-employment in traditional face-to-face survey data collection that allows for proxy respondents and non-private interviews – in comparison to interviewing adult household members in private. However, the effects are less pronounced for 7-day recall (the reference period used in phone surveys) versus 12-month recall. 

Ultimately, in our set-up, however, it would be speculative to quantify potential proxy response bias. We argue that proxy response is second-best to self-reporting of all individuals but that it is still preferable to have data on all (adult) individuals, even if reported through a proxy. Given the magnitude of the differences between respondents and the general adult population, we contend that errors stemming from proxy reporting could be a second-order concern. As such, we feel as long as we acknowledge the issue of proxy response adequately, it is worthwhile to present the results in sections 2.6 and 3.3 for illustrative purposes. 

Comment: Finally, with the large set of variables that the authors look into, times the 4 countries for which the analyses are replicated, there are a lot of numbers to review in order to draw conclusions. Although I see why the weights are estimated at the country level, in presenting the results on bias correction, the regressions could be presented at the aggregate level, combining all countries in one regression, and the country-specific analyses could be presented in an appendix. Alternatively, would there be scope for an aggregation over the different variables, in order to reduce the number of coefficients that need to be interpreted and aggregated by eyeballing before drawing conclusions?

Response: The volume of numbers and figures to look at is indeed large. However, we feel it is important to keep the analysis at the country-level to allow for an evaluation of how the country-specific contact and selection protocols may have affected the comparisons between the respondents and the general adult population and the success of our bias correction. 

A case in point is the Malawi survey whose particular contact protocol we now discuss in greater detail (in response to your second comment). We further hope that once the relevant figures are next to the text in the final published manuscript, it will be possible to draw conclusions from the results more easily. Similarly, we do not believe it would benefit the analysis to aggregate over different variables, as most of them are binary based on underlying categorical variables. 

Comment: Other comments:

- The first part of the introduction appears negligent of a large body of literature on phone surveys and survey design; but in fact this literature is mentioned towards the end of the introduction. In this case, I would consider integrating the existing literature in the first part of the introduction and presenting the findings along with their implication as the key contribution, to shorten the introduction and appear less negligent of the work that has already been done in this field.

Response: Thank you for this comment. We have integrated the relevant references from the existing literature early on the in the Introduction, which we have also refocused and shortened in line with your first comment. We have opted to retain a short paragraph summarizing this literature at the end of the Introduction in an attempt to do justice to the breadth of the literature, some of which is not directly related to the problem we analyze but still worth mentioning in the paper for readers who would like to dive deeper into related issues.

Comment: The figures were not visible in the manuscript itself, only as the appendices of the submission.

Response: Apologies for this, this was based on our interpretation of the PLOSONE required formatting. We understand in the final published article, the figures will again appear alongside the text to which they pertain.

Comment: A bit more explanation on why certain things vary across countries would be useful. For instance, why are the fifth round data for two countries excluded? Did these not include the employment outcomes? But how does that rhyme with all surveys being standardized across the four countries? And why were the selection protocols different in the four countries?

Response: We have clarified this in the text in section 2.1: questionnaire design was comparable and a questionnaire working group developed modules which countries could then adopt. A set of core modules was adopted by all countries while other, optional modules were adopted according to country needs and interest. This is why certain topics are covered in some countries but not in others, including the fifth-round data on individual-level employment outcomes. We added an additional sentence in section 2.1 explaining that the individual-level employment data was collected only in these two countries. 

As for selection protocols and interview protocols more broadly, these followed the advice provided in Amankwah et al. (2020). At the same time, as these are NSO-owned and World Bank-supported surveys, there is always scope for customization and contextualization and selection protocols reflected what NSOs and World Bank support staff jointly considered to lead to the best outcomes in terms of high response rates and data quality. Another factor was that each of the face to face baseline surveys recorded household contact phone numbers in a slightly different way – ranging from only collecting the phone number for the head of household to collecting the phone numbers for all household members. This then had a bearing on contact protocols and also likely affected the selection of the main respondent. Part of the implications of this paper is that in future survey implementers need to think more critically about recording and curating contact phone numbers and about selection protocols than was done with operational urgency when the COVID-19 phone surveys were set up. In any case, we have reflected these points in the manuscript in section 2.3.

Comment: At times, the paper reads a bit like a promotional brochure for the World Bank (for instance the introduction claims that the World Bank is the prime institute doing and learning around phone surveys, followed by 3-4 other institutions, among others). The fact that the anonymized surveys are published online a few weeks after completion could be a footnote in the methods section and does not require an entire paragraph. The paragraph in the conclusion stating how widely these data are used is irrelevant unless it is used to illustrate that people are using the data for individual-level analyses and therefore using the wrong household-level weights.

Response: Thank you for this comment. Since the PLOSONE style guide does not allow for footnotes, we had initially moved the information on publication of the surveys after a few weeks into the body text. However, we have now removed that information, retaining only what we considered relevant to data collection methodology. The idea of the paragraph in question in the conclusion was to highlight that there is now quite a lot of phone survey data out and that it is being used widely – and so our findings pertain to a growing and increasingly used data type. Arguably, the paragraph did not communicate that idea clearly enough and we have reframed it, to expand beyond just the four countries question. 

Comment: Bottom of page 19 (last sentence), note that there is a small typo: households instead of household.

Response: Thank you. We have fixed this.

Comment: IRB: Clarify that the data to which the authors had access were anonymized and that the merging of phone survey and face-to-face data was based on an anonymized household ID. Even if for data collection no ethics approval was needed, for a researcher to work with data that contain personal identifiers, and merge data from different sources, IRB should be obtained.

Response: Thank you for pointing this out. We clarified that the data were anonymized/de-identified and the household and individual IDs were likewise anonymized.

Comment: Overall, though, the paper presents an extremely useful analysis that needs to be published, since we should be more aware that our household-level weight adjustments often do more harm than good if using individual-level data, and that survey respondent selection protocols need to be adjusted when the objective is individual-level analysis.

Response: Thank you very much.

References

Henderson, S., Rosenbaum, M., 2020. Remote Surveying in a Pandemic: Research Synthesis. Innovation for Poverty Action.

Himelein, K., 2014. Weight Calculations for Panel Surveys with Subsampling and Split-off Tracking. Statistics and Public Policy 1, 40–45. https://doi.org/10.1080/2330443X.2013.856170

Himelein, K., Eckman, S., Kastelic, J., McGee, K., Wild, M., Yoshida, N., Hoogeveen, J., 2020. High Frequency Mobile Phone Surveys of Households to Assess the Impacts of COVID-19. Guidelines on Sampling Design. World Bank, Washington D.C.

Little, R.J.A., Lewitzky, S., Heeringa, S., Lepkowski, J., Kessler, R.C., 1997. Assessment of weighting methodology for the National Comorbidity Survey. American journal of epidemiology 146, 439–449.

Rosenbaum, P.R., Rubin, D.B., 1984. Reducing Bias in Observational Studies Using Subclassification on the Propensity Score. Journal of the American Statistical Association 79, 516–524. https://doi.org/10.2307/2288398

---

## [Decision Letter · Decision Letter 1]

8 Oct 2021

Representativeness of individual-level data in COVID-19 phone surveys: Findings from Sub-Saharan Africa

PONE-D-21-17113R1

Dear Philip,

I heard back from the two reviewers and they both indicated that all their comments and suggestions were satisfactory addressed. Therefore, I have decided to accept the article as is. Congratulations!

Kind regards,

Bjorn Van Campenhout, Ph.D.

Academic Editor

PLOS ONE

---

## [Editor Report · Acceptance letter]

14 Oct 2021

PONE-D-21-17113R1 

Representativeness of individual-level data in COVID-19 phone surveys: Findings from Sub-Saharan Africa 

Dear Dr. Wollburg:

I'm pleased to inform you that your manuscript has been deemed suitable for publication in PLOS ONE. Congratulations! Your manuscript is now with our production department. 

Kind regards, 

on behalf of

Dr. Bjorn Van Campenhout 

Academic Editor

PLOS ONE